# Functional and Nutritional Characteristics of Natural or Modified Wheat Bran Non-Starch Polysaccharides: A Literature Review

**DOI:** 10.3390/foods12142693

**Published:** 2023-07-13

**Authors:** Wendy Sztupecki, Larbi Rhazi, Flore Depeint, Thierry Aussenac

**Affiliations:** Institut Polytechnique Unilasalle, Université d’Artois, ULR 7519, 60026 Beauvais, France; wendy.sztupecki@unilasalle.fr (W.S.); larbi.rhazi@unilasalle.fr (L.R.); flore.depeint@unilasalle.fr (F.D.)

**Keywords:** dietary fibre, arabinoxylans, β-glucans, health effects, characterisation, modification

## Abstract

Wheat bran (WB) consists mainly of different histological cell layers (pericarp, testa, hyaline layer and aleurone). WB contains large quantities of non-starch polysaccharides (NSP), including arabinoxylans (AX) and β-glucans. These dietary fibres have long been studied for their health effects on management and prevention of cardiovascular diseases, cholesterol, obesity, type-2 diabetes, and cancer. NSP benefits depend on their dose and molecular characteristics, including concentration, viscosity, molecular weight, and linked-polyphenols bioavailability. Given the positive health effects of WB, its incorporation in different food products is steadily increasing. However, the rheological, organoleptic and other problems associated with WB integration are numerous. Biological, physical, chemical and combined methods have been developed to optimise and modify NSP molecular characteristics. Most of these techniques aimed to potentially improve food processing, nutritional and health benefits. In this review, the physicochemical, molecular and functional properties of modified and unmodified WB are highlighted and explored. Up-to-date research findings from the clinical trials on mechanisms that WB have and their effects on health markers are critically reviewed. The review points out the lack of research using WB or purified WB fibre components in randomized, controlled clinical trials.

## 1. Introduction

Cereal-based products have always been a part of the human diet as a source of energy and nutrients. According to the Food and Agricultural Organization (FAO), the forecasted world cereals utilisation for 2022/2023 is around 2778 million tons [1]. Among cereals, human consumption of wheat is increasing. U.S. Wheat Associates reports an increase by 90 million metric tons between 2008 and 2019 [2]. The human cereal diet mostly consists of refined grain products. Indeed, starch, the main component of cereal grain, is an important energy source, but very poor in dietary fibres (DF) and related bioactive compounds. 

Hipsley (1953) was the first to use the term dietary fibre to describe a nutritional property of diets [3]. The Codex Alimentarius Commission (CAC), a FAO/World Health Organization (WHO) body, spent almost twenty years elaborating a comprehensive definition of dietary fibre which would be recognized by the entire international community. In June 2009, a definition of dietary fibre was adopted by the CAC [4].

A consensus on definition states that dietary fibre means carbohydrate polymers with at least 10 monomeric units, which are not hydrolysed by the endogenous enzymes in the small intestine of humans. However, inclusion of resistant oligosaccharides with three to nine monomeric units is left to the discretion of the national authorities. Thus, these oligosaccharides are considered as dietary fibre by the European Food Safety Authority (EFSA) [5], the U.S. Food and Drug Administration (FDA) [6], Health Canada [7] and Food Standards Australia and New Zealand (FSANZ) [8].

The dietary fibre definition concerns three kinds of carbohydrates: naturally occurring fibres, carbohydrate polymers obtained by physical, chemical or enzymatic treatments and synthetic carbohydrate polymers. Beneficial physiological effects such as laxation, cholesterol reduction, glucose reduction and increasing of insulin sensitivity [9], which are scientifically proven, are also considered.

Most definitions include non-starch polysaccharides, resistant oligosaccharides, and resistant starch. Associated substances such as lignin and other non-glucuronic compounds that are linked to cell wall polysaccharides are also included [10].

Cereals are the main source of human fibre intake, providing 50% of the fibre contribution, from vegetables (30–40%), fruits (16%), and nuts 3% [11]. Among cereals, wheat is rich in these non-starch polysaccharides, which are situated mostly in the bran (Figure 1). Wheat Bran (WB) is a by-product of wheat flour milling. It consists of approximately 50% dietary fibre. During the crushing or breaking stage, germ and bran are separated from the endosperm. This is an important goal of the miller. The bran fraction is sifted out to improve dough-making properties and end-product sensory qualities. The addition of WB makes processing wheat flour more difficult and decreases the organoleptic qualities, making the fibre-rich products less attractive to the consumers [12]. Although wheat bran is rich in phytochemicals, micronutrients, bioactive compounds and fibres, it is rarely valorised in human nutrition and is mostly used to feed livestock.

It is known that a high-fibre diet provides significant health benefits. Epidemiological reports suggest beneficial effects on type-2 diabetes (T2D), obesity, colon cancer prevention, and reduction of cardiovascular diseases (CVD) with a high-dietary-fibre diet [13,14,15,16,17]. In addition, nutritional claims such as “high fibre” engage consumers to eat a healthier and more high-fibre diet [18]. In WB, these prominent health benefits are mostly attributed to the high content of DF, their structure and the bioavailability of associated compounds (i.e., polyphenols) [10,19,20,21,22,23].

**Figure 1 foods-12-02693-f001:**
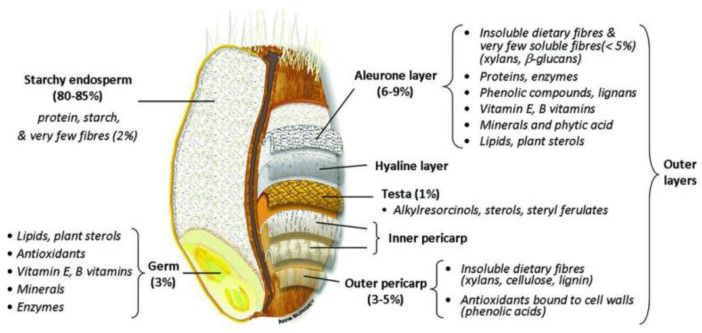
Schema of wheat grain anatomy [14].

Manufacturers and researchers try to optimize and improve the properties of bran dietary fibre for a variety of functional food products with added value. In addition, the goal of the milling industry and food researchers became to improve the nutritional and health profile of WB while facilitating whole-wheat flour process. For this purpose, some processes have been developed on WB to modify the functional properties of DF using biological, physical, chemical, or combined treatments [24,25]. Although the physico-chemical modifications on WB have been described, few studies investigate the biological effects of modified WB on health. 

Despite these growing concerns over WB-DF, there have been no or very few in vivo randomized controlled trials in humans assessing the health benefits of WB, to the best of our knowledge. No review has provided a unique and comprehensive overview regarding the WB functional properties, process modifications and randomized, controlled clinical trials. In addition, the mechanisms that regulate the relationship between WB/DF and their beneficial health effects are not fully understood. Thus, in this review, after shortly providing WB composition, we will attempt to explain clearly how fibre molecular structure (i.e., composition, distribution, content, molar masses, etc.) is linked to its physico-chemical properties. Then, depending on the availability of clinical studies, we will report the health effects of WB by considering structural and functional properties of wheat bran components. Finally, we will focus on existing WB modifying processes and whether the assumptions on modified-WB improvement of nutritional and health effects are validated or not.

## 2. Wheat Bran Characteristics and Properties

### 2.1. Chemical Composition of Wheat Bran

Wheat grain is composed of the germ and the starchy endosperm which is the energetic reserve for germination. They are surrounded by a series of layers from the aleurone to the pericarp (Figure 1). In the milling industry, these layers are called “wheat bran” (WB), and commercial wheat bran is mainly composed of these outer layer remnants of starchy endosperm. The bran accounts for 14 to 19% of wheat grain [26]. Its exact composition depends largely on the milling process. Wheat bran typically contains 43 to 60% (*w*/*w* dry matter) NSP, 11–24% starch, 14–20% protein, 3–4% lipids, and 3–8% minerals [27,28,29,30] (Table 1). 

#### 2.1.1. Wheat Bran Fibre Carbohydrates

The fibre carbohydrates are classified according to their structure: NSP have 10 or more sugar monomer units (hemicelluloses, cellulose, pectin), while resistant or non-digestible oligosaccharides have less than 10 units such as fructans. Fibre classification depends also on their solubility in water, or fermentability [31]. Among wheat bran NSP, arabinoxylan is the most represented (70%), cellulose accounting for 19%, and β-glucans for 6% [32]. Hence, the amount of AX found in wheat bran range from 5 to 27% of bran [33,34,35,36,37,38] (Table 1). It should be noted that wheat composition varies depending on the genotype and environment, so the choice of the wheat variety is important for the optimisation of health and technological parameters [39].

Hemicelluloses are comprised of glucans, xylans (arabino-, glucurono-, glucurono-arabino-xylans), mannans, and β-glucans [10]. They are heteropolysaccharides connecting cellulose to lignin [40]. WB contains 30% to 39% hemicellulose [41,42] originating from the pericarp and the aleurone layer, where polysaccharides represent 25% of the total NSP. AXs, also called hemicellulose B, are the major constituents of hemicellulose, accounting for 10.9% to 26% [43,44]. They are built from xylose, arabinose and pentose sugars. AXs consist of a linear backbone of β-(1–4)-D-xylopyranose subunits to which α-L-arabinofuranose residues are attached on position O-2 and/or O-3 [30] (Figure 2). The arabinose/xylose (A/X) ratio can be used as indicator for AX structure, solubility, and localisation. It varies from 0.2 to 1.1 (Table 2), the AX in the outer layers being less substituted by arabinose that those of the inner endosperm [45]. 

β-glucans in wheat are homopolymers of D-glucopyranosyl with 2 or 3 β-(1→4) linkages separated with 1 β-(1→3) linkage. The content of β-glucan is less than 3% in dry matter in wheat bran [47,48] (Table 1). The degree of polymerisation (DP) varies between 5 and 28. β-glucans can be hydrolysed into trisaccharides (DP3) and tetrasaccharides (DP4) which, together, constitute more than 90% of β-glucan structure, and some oligosaccharide fragments [49] (Figure 3). β-glucans of cereals show structural similarities, but the molar ratio of cellotriose to cellotetraose units (DP3/DP4) varies in wheat (3.0–4.5), barley (1.8–3.5), and oat (1.5–2.3) [50,51].

Cellulose is a structural polysaccharide located in the outer pericarp layer of the grain. It represents 10 to 33% of WB (Table 1) [41,42]. It is a glucose homopolymer similar to starch, but cellulose is β-linked [52]. Pectin is linked with cellulose and hemicellulose in the cell walls. They are composed of rhamnogalacturonan I and II, xylogalacturonan, and homogalacturonan [53], the latter representing about 65% of pectin molecules. Arabinogalactan is a soluble pectin [53,54]. The major simple units which constitute these domains are: galacturonic acid, rhamnose, galactose, and arabinose [55]. They have viscous or gel-forming capacities [10]. 

**Table 1 foods-12-02693-t001:** Composition of total, soluble, and insoluble fibre from wheat bran and example of methods used in the bibliography.

Molecular Group	Most Used Method in Literature	Values in Total WB	Values in WB SDF	Values in WB IDF	References
DF (g/100 g WB DW)	Gravimetric-enzymatic (AOAC methods (991.43; 985.29), AACC methods (32-07.01; 32-05.01; 32-21.01; 32-06.01))	33.4–62.4	2.3–9.8	38.5–60.1	[28,37,56,57,58,59,60]
Total starch (g/100 g DW)	Colourimetric methods (AOAC 996.11, AACC method 76–12)	11.3–23.5	X	7.6	[37,57,58,59,60,61,62,63]
Nitrogen (g proteins/100 g DW)	Kjeldahl method (AACC 46-10) combustion methods (Dumas ICC standard 167)	14.5–20.9	X	9.9	[59,60,63,64,65,66]
Simple sugars (g/100 g WB DW)	Chromatography (HPLC-ELSD, HPAEC)	0.14–0.63	X	X	[67,68]
Ash (g ash/100 g DW)	Gravimetric (AACC-08-01)	4.4–6.4	X	3.9	[58,62,63]
Cellulose (% DW)		5.5–31.1	X	X	[41,42,69]
Lignin content	Gravimetric (Klason)	8–15	X	X	[41,42,54]
Total β-glucans	AACC 32-23.01 AOAC 995.16	2.1–2.3	1.9	2.3	[62,70]
Total arabinoxylans (% of DW)	Colourimetry (phloroglucinol), Chromatography (GC-MS)	5.0–26.9	X	X	[36,37,38]

American Association for Cereal Chemistry (AACC) and Association of Official Analytical Chemists (AOAC) reference methods. 

The nitrogen content analysis serves to determine protein content due to a coefficient of 6.25 in the general method and is found in the bibliography, but 5.7 is more adapted for wheat. The ash content is relative to the mineral quantification.

**Table 2 foods-12-02693-t002:** Structure and functional properties of wheat bran compounds and examples of methods used in the bibliography.

Physico-Chemical Property	Methods	Value in Wheat Bran	References
WEAX (g/g DW)	Chromatography	0.65	[70]
Molecular Bounds	Spectroscopy FTIR	3300 cm^−1^: O-H (cellulose, hemicellulose) 2930 cm^−1^ C-H CH_2_ (polysaccharides) 1660 cm^−1^ (lignin)	[63,71,72,73]
Ratio Arabinose/Xylose	Chromatography (HPLC-ELSD, HPAEC-PAD)	0.2–1.1	[74,75,76,77]
AX Solubility (%)	Gravimetric (extractions)	15	
AX Molecular weight (kDa)	Asymmetrical Flow Field-Flow Fractionation (AFFFF)	20–600	[54,75,78]
β-glucans Mw (kDa)	AFFFF	258–635	[49,51,79]
β-glucans Polymerisation degree (DP)	Enzymatic and Chromatographic (HPLC)	5–28	

#### 2.1.2. Wheat Bran Fibre-Associated Phenolic Compounds

The bran layers (aleurone layer, nucellar epidermis, inner pericarp, and outer pericarp) contain phenolic acids that are mostly cross-linked with cell wall structural polysaccharides through ester bonds [80,81,82]. Wheat bran contains between 1.4 and 11.1 mg of gallic acid equivalent (GAE) of total polyphenols per g of dry weight (DW) [25,44] (Table 3). The content of phenolic acids is shown to be affected by genetic factors, as well as by environmental interactions, which leads to a large variation [83,84,85]. 

Wheat bran polyphenolic molecules are mostly composed of derivatives of hydroxybenzoic acids, hydroxycinnamic acids, and flavonoids [44,86] (Table 3). The most abundant compounds are ferulic acid (FA), dehydrodimers and dehydrotrimers of ferulic acid, and sinapic and *p*-coumaric acids belonging to derivatives of hydroxycinnamic acids [83,84,87,88]. Polyphenols as ferulic and *p*-coumaric acids are linked with AX at the C-5 position of arabinose [22,44]. FA is mostly bound to AX (98.8%). However, a small fraction of FA remains in freely soluble (0.2%) and soluble conjugated form (1%) [89]. Arabinogalactan is also rich in FA [53,54]. FA mainly cross-link polysaccharides, whereas *p*-coumaric acid would also cross-link lignin. Ferulic acid constitutes about 0.5% *w*/*w* of wheat bran [61]. 

Lignins are phenolic polymers consisting of three monolignol units: guaiacyl, syringyl, and *p*-hydroxyphenyl. They are linked with hemicellulose thanks to polyphenols. FA ester-linked to glucuronoarabinoxylan are nucleation sites for lignin polymerisation through ester bounds anchoring lignin to the polysaccharide moiety [90]. These non-carbohydrate aromatic polymers represent 8 to 15% of the wheat bran [41,42,54] (Table 1). 

Alkylresorcinols are not considered fibre components [10]. They are phenolic lipids counting for 2.7 mg/g in wheat bran and varying between 489 and 1429 µg/g DW in wheat whole grain [91,92,93]. Alkylresorcinols are amphiphilic 1,3-dihydroxybenzene derivatives in which an alk(en)yl chain of 15 to 25 carbon atoms is attached to C5 of the benzene ring [94]. They play an important role in health and nutrition benefits of wheat bran, and their effects are close to those of fibres, namely antioxidant, antimicrobial and anti-inflammatory properties.

**Table 3 foods-12-02693-t003:** Polyphenol contents in wheat bran.

Polyphenol	Quantity µg/g of Wheat Bran DW	References
Polyphenol		
Free Phenolic Compounds		
Phenolic acid—Hydroxybenzoic acids		
Gallic acid	0.3–1.0	[44,58]
Vanillic	2.2–28.5	[58,95,96]
Vanillic acid isomer	8.1–15.5	[97]
4-hydroxybenzoic = *p*-hydroxybenzoic acid	2.5–5.8	[58,96]
Protocathechuic	Nd–8.9	[44,96]
Syringic	1–8.0	[44,95,96]
Salicylic	6.4	[96]
Ellagic acid		[98]
Phenolic acid—Hydroxycinnamic acids		
3.4- Dimethoxycinnamic	1.6	[96]
Caffeic acid	0.1–1.6	[44,58,96]
*Trans*-caffeic	2.5–8.2	[97]
Chlorogenic acid	1.9–3.1	[44,58]
*p*-coumaric	10.6–50.2	[58,96]
*Trans-p*-coumaric	1.1–2.3	[97]
Ferulic	Free: 0.2–19Total: 1375–5670	[44,58,95,99]
*Trans*-ferulic	8.4–20.2	[96]
*Cis*-ferulic	Nd–0.7	[96]
Sinapic	1.6–5.8	[44,58,96]
Phenolic acid—Phenylethanoid		
Hydroxytyrosol	5.6–12.4	[97]
Flavonoids		
Catechin	3.5–50.1	[44]
Epicatechin	1.1–3.3	[44]
Apigenin-6-C-arabinose-8-C-hexoside	101–149	[97]
Apigenin-6-C-β-galactosyl-8-C-β-glucosyl-O-glucopyranoside	37.9–49.3	[97]
Apigenin-6,8-di-C-glucoside	3.4–6.5	[97]
Anthocyanins		
Malvidin		[98]
Total contents		
Total phenolics (FC) GAE	1467–11,100	[30,60,69]
Bound polyphenols GAE	2451–8500	[60]
Free polyphenols GAE	1175–2600	[60]

### 2.2. WB Physico-Chemical Properties

As seen above, WB contain a wide array of molecules. Due to their complex and diverse structure, fibre gives certain physico-chemical properties to wheat bran, of which solubility, viscosity, and binding properties are not always well defined in the literature. Fibre-associated compounds such as polyphenols bring also physico-chemical bioactivity to wheat bran.

#### 2.2.1. Fibre Functional Properties

##### Wheat Bran Fibre Solubility

According to solubility in water, DFs could be classified as soluble dietary fibre (SDF) or insoluble dietary fibre (IDF). The sum of IDF and SDF amounts gives total dietary fibre (TDF) content. Wheat bran contains about 50% (*w*/*w*) TDF, and more than 90% of these fibres are water-insoluble [14] (Table 1). WB IDF consists of hemicellulose (water-insoluble AX), cellulose, lignin, and resistant starch, while SDF are oligosaccharides and soluble hemicelluloses: water-soluble AX, galactomannans, and β-glucans. Twenty-five to fifty % of AX are soluble in water (WEAX) and they differ from insoluble AX with their higher substitution degree and higher heterogeneity. The solubility of AX polymers is influenced by the degree of substitution and the distribution pattern of arabinofuranose residues [100,101]. The FA crosslinks also influence the water solubility of the AX molecules. The solubility decreases by covalent cross-linking between chains due to FA dimerisation. It has been shown that alkaline extraction improves AX’s solubility by removing these ester-linked FA [102]. The solubility of β-glucans is influenced by the distribution of β-(1.3)- and β-(1.4)-linkages [11]. The DP3:DP4 ratio is tightly correlated to the relative solubility of the β-glucan [103], which is known to be significantly controlled by growing location, cultivar, and the location–cultivar interaction, as well as growing-year factors [104].

IDF, SDF and TDF contents of wheat bran samples are measured by various methods. Although some methods are standardized, DF (SDF, IDF and TDF) contents are method-dependent and thus are subject to sharp and pronounced variation. Non-enzymatic-gravimetric (NEGM) [105], enzymatic–gravimetric (EGM) [106], and enzymatic–chemical (colourimetric and chromatographic) methods (ECM) [107] are the principal analytic methods for fibre content determination. The EGM shows the highest content because it estimates polysaccharides, lignin, resistant starch, non-digestible oligosaccharides and others (waxes, phenolic compounds, Maillard reaction products) [108]. NEGM underestimates TDF content because water-soluble components are not measured, and ECM shows the lowest content of fibre due to the loss of polysaccharides during hydrolysis [108]. TDF content analysis in wheat is currently determined by official AOAC methods (991.43; 985.29) and AACC methods (32–07.01; 32–05.01; 32–21.01; 32–06.01) [109]. AOAC 991.43 and 985.29 methods do not allow correct measurement of all parts of resistant starch and determine little or none of the non-digestible oligosaccharides [110]. That is why AOAC 2009.01 and 2011.25 methods have been set up to quantify all types of resistant starch and all oligosaccharides including the low Mw fibres with DP 3–9 [110,111]. These methods are based on enzymatic hydrolysis and gravimetric analysis. Enzymes currently used are: α-amylase for gelatinisation, hydrolysis, and depolymerisation of starch; protease to solubilise and depolymerise proteins; and amyloglucosidase to hydrolyse starch fragments to glucose. Each enzyme needs a specific condition (pH, temperature, incubation time) to be activated. A strong correlation between AX and TDF contents in wheat grain-based products has been observed [38]. Thus, measuring the AX content can be used to estimate TDF content in WB.

In the absence of tangible and reliable data based on clinical studies to predict and clearly explain physiological effects by DF solubility, FAO/WHO have recommended the phasing out of the distinction between soluble and insoluble fibres. However, this solubility-based classification is still widely used, as SDFs constituted of non-cellulosic polysaccharides (e.g., pectin, gums, psyllium, mucilage) are being increasingly used and studied. In addition, these water-soluble fibres are showing effectively physiological effects.

##### Wheat Bran Viscosity

Viscosity is an important criterion, which is influenced by the fundamental molecular structure (conformation, molecular weight, molecular weight distribution) of DF polymers, their solubility and concentration [112,113,114]. In wheat, a quantitative trait locus (QTL) has been found to explain 32–37% of the variation in the relative viscosity [115]. AXs and β-glucans, due to their highly asymmetrical conformation, form a highly viscous solution [11]. Both WEAX and insoluble AX form highly viscous solutions displaying pseudoplastic behaviour. The AX’s viscosity solutions are controlled by AX chain length and substitution degree. They are also determined by the AX substitution pattern. The viscosity depends on the high molecular weight (Mw), stiff, semi-flexible random coil conformation, molecular weight distribution, and concentration of both WEAX and Alkali Extractible AX (AEAX). These parameters, in particular Mw, are influenced by the localisation of AXs in the wheat grain tissue and genotype [33]. FA cross-linked to AXs are determinant in controlling the viscosity, by increasing the Mw and modifying the AX conformation. A higher viscosity is observed with higher content and Mw of AX molecules [19,116]. The viscosity of β-glucan dispersions is shown to be influenced by cultivar factor, location and shear rate [104]. The asymmetrical shape is considered responsible for the high viscosity of β-glucans. It is also related to β-glucan concentration, which is shown to be influenced by genotype factors in oats [103]. The viscosity of the β-glucans is suggested to be dependent on the molar mass, concentration, and ability to form aggregates [117]. Since the intrinsic viscosity is influenced by the Mw of dietary fibre components and because the molar mass of β-glucan molecules is higher than that of AX polymers, β-glucan will have more impact on viscosity than AX.

##### Binding Capacities

Thanks to their structure and composition, fibres have binding properties to different molecules or patterns. Water, oil, glucose, cholesterol, cation exchange, nitrite, or sodium cholate binding capacities are studied.

##### Water Holding Capacity and Water Swelling Capacity

Water holding capacity (WHC) is the amount of water that can be absorbed by WB while water swelling capacity (WSC) corresponds to the volume occupied by the swollen bran after WB hydration [118]. These properties could be considered as useful parameters for predicting the faecal bulking ability of a DF source [119]. WHC can be determined by Baumann apparatus or by centrifugation. In the absence of any standardised method, variations can be partly attributed to the methods used. Thus, the amount of water is generally higher for the centrifugation method [108]. WHC ranges between 2.2 and 7.3 g/g WB DW (Table 4). WHC depends on the content of IDF, the intact cell structure of bran [120]. AXs are capable of absorbing an important quantity of water (4 to 10 g/g DW) [121]. Water Unextractable AX (WUAX) are able to absorb 7- to 10-fold their dry weight while WEAX have a lower WHC, about 4–6 times their dry weight [122]. Other authors found that the water-insoluble fraction of pentosans (including AX) shows a WHC of about 10-fold their dry weight, while the water-soluble faction has 11-fold their dry weight [26,123]. This means that absorption is dependent on the pentosane type and composition.

At the molecular level, variations in water absorption might be explained by differences in the AX structure. The conformation and the molecular dispersity of AXs in wheat depends on the length of the xylan backbone, the A/X ratio, the xylan polymerisation, the substitution and the distribution pattern, and the linking of ferulic acids to other molecules of AX [121,124]. AXs containing a high FA content enhance water-binding capacity (WBC), since these fractions of AX were found to form an extensive crosslinking system, leading to well-developed gel networks [125].

β-glucan molecules absorb more water than WEAX and WUAX. β-glucan WHC is related to the very hydrophilic features of β-glucan due to the abundance of hydroxyl groups that form hydrogen bonds with water and give the molecule a capacity to absorb water [126,127]. The β-(1→3) linkage makes the molecule flexible and can explain the high water-binding capacity of β-glucan.

WHC can also be affected by particle-size distribution, surface area, and porosity on DF [72]. The hydration capacities of DFs are mainly related to the porous complex structure containing many functional groups, which could retain water molecules through hydrogen bonds, considering the thermodynamics and dynamics of water absorption/desorption phenomena [128]. The number and nature of its water binding sites plays a role in water retention [119]. 

##### Oil-Binding Capacity

Oil-binding capacity (OBC) is also called oil-holding capacity (OHC), depending on whether it is expressed in mL/g or g/g [129]. OBC is the amount of vegetable oil retained by WB and reflects the ability of WB components (mainly DFs) to absorb and retain fat and to interact with lipids. The OHC is found to be related to fibre particle surface characteristics, overall charge density, and the hydrophilicity of constituent polysaccharides [108]. WB OBC is influenced by the bran’s particle size. This parameter decreases when the mean of particle size distribution decreases [130]. As the specific surface area increases with the reduction of particle size, OBC would most likely depend on the porosity of the DF structure rather than the affinity of the DF molecule for oil. However, except for these physical factors, OBC seems to be also influenced by the presence of lipophilic sites, overall hydrophobic property, hydrophilic nature, and capillary motion [131]. The method used consists of mixing WB with oil and recovering pellets after centrifugation. Variations of OBC have been observed in the literature (Table 4). They could be attributed to the protocol, which is not normalised and seems to be less reproducible.

##### Glucose Adsorption Capacity

Glucose adsorption capacity (GAC) is the ability of WB to bind and retain glucose. It is a useful in vitro parameter to evaluate the effects of WB fibre on the absorption of glucose in the gastrointestinal tract. GAC relates DF physical properties to glucose metabolism because glucose is adsorbed by DF, thereby reducing contact with the human intestinal tract [132]. The GAC can be attributed to the total dietary fibre content [133,134,135,136]. It is, however, positively influenced by the SDF content because the high viscosity of SDF delays glucose molecule absorption in the gastrointestinal tract [137]. SDF exhibits more affinity for glucose than IDF [63,71]. This affinity is improved with increasing concentration of glucose (Table 4). GAC is also positively affected by DF structural changes, such as reduced particle size distribution which is always accompanied by an increase in the specific surface area [138]. In addition, a more porous DF network would allow the glucose molecules to be incorporated into their binding sites in bran particles and likely improves the interaction between DF in bran and molecules of glucose.

To characterise WB capacity to reduce glucose absorption, α-amylase inhibition capacity is also measured. α-amylase is a key enzyme in the digestion of starch in the human body. SDF with lower Mw and viscosity shows a stronger inhibitory effect on α-amylase [139]. 

##### Cholesterol Adsorption Capacity

Cholesterol adsorption capacity (CAC) is the ability of wheat bran to bind cholesterol molecules. It is determined by a colourimetric method which serves to assess the adsorption of lipophilic substances by the sample. The pH seems to play a role in the CAC [63]; CAC is around four times lower at pH 2 compared to pH 7 (Table 4). Data suggest that interactions between DF and cholesterol molecules are not based on ionic bonds. Both are positively charged in acidic environments, leading to repulsion phenomena. Neutral pH corresponds to the small intestine environment and acidic pH to the stomach environment. This indicates that WB can potentially adsorb more cholesterol in the small intestine than in the stomach. There is a positive correlation between SDF content and higher ability to adsorb cholesterol [140]. Therefore, the CAC property can prevent CVD by reducing cholesterol uptake from the diet.

##### Sodium Cholate Adsorption Capacity

Over 90% of human bile acids exist in binding form, such as sodium cholate. DF can adsorb intestinal sodium cholate and lipid substances [141]. In addition, cholesterol will be converted by the body into sodium cholate to regulate the loss, thereby promoting the consumption of endogenous cholesterol [141]. The SCAC and the CAC play a role in lowering blood pressure and blood lipid. Sodium cholate adsorption capacity (SCAC) increases with IDF content [142] (Table 4). Interactions between bile acids and DF are not totally understood. SCAC varies with bile acids and DF types because of their different physico-chemical characteristics (ex: particle size, surface area, substitution rate). The viscous capacity of DF is also a proposed mechanism [143]. 

The sodium cholate adsorption capacity of TDF is around 31% in WB with cholestyramine standard, which represents 100% bound. There is not an official method for SCAC analysis. Most articles use a method inspired by Camire et al. (1993) [144], modified by Kahlon and Chow (2000) [145]. First, an acidic incubation is done in the sample with HCl to simulate gastric digestion; then, the pH is adjusted at 6.3 to reproduce the in vivo condition in the human duodenum [145]. The sample is finally incubated with sodium cholate and the unbound cholate in the supernatant is read with a spectrophotometer [146]. In the oldest methods, porcine enzymes were added to stimulate small intestine conditions [144,145]. 

##### Cation Exchange Capacity

Cation exchange capacity (CEC) is the capacity for DF to retain cations due to carboxyl and hydroxyl side groups. This parameter is interesting due to the cation exchange phenomenon, which is known to increase binding of heavy metals and reduce adsorption of cholesterol. Correlations between CEC and high-density lipoprotein (HDL)/total cholesterol ratio and also with uronic acid contents of legume seeds IDF have been reported [147]. The CEC adsorption can have undesired effects through the reduction of essential cations. CEC value is inversely proportional to the pH value [142].

##### Nitrite Ion Adsorption Capacity

Dietary fibre has the ability to trap minerals and limit their bioavailability by shortening the transit time, which reduces the absorption time of minerals by the body. This can also trap minerals physically by electrostatic binding [148].

This property can be an advantage. Nitrites are toxic compounds for animals, and risks can be prevented through the nitrite adsorption capacity of wheat DF. For nitrite ion adsorption capacity (NIAC) analysis, a fibre sample is mixed with NaNO_2_ at pH 2. The residual nitrite ion concentration is measured by spectrophotometry, or chromatography, or the naphthalene ethylenediamine hydrochloride method [149,150]. NIAC depends on pH and phenolic acids. Indeed, in acidic conditions (pH 2), FA and lignin react with nitrite ions. When pH increases, the carboxyl group of phenolic acids dissociates and increases negative charge density, leading to release of nitrite ions and reducing NIAC by more than two times [73]. 

**Table 4 foods-12-02693-t004:** Functional properties of wheat bran, WB SDF and WB IDF according to the literature.

Functional Property	Value in WB DW	Value in WB SDF	Value in WB IDF	References
Water holding capacity (g water/g DW)	2.2–7.3	2.2	3.0–4.3	[37,56,59,60,62,63,70,71,73,108,130,142,151,152,153,154]
WSC (mL/g)	1.7–2.7			[59,142]
Oil holding/retention capacity (g oil/g DW)	2.3–2.5	0.3–1.5	1.5–3.5	[59,63,73,86,154]
Oil binding capacity (mL oil/g DW)	1.2–5			[130,155]
Cholesterol adsorption capacity (mg/g)	2.2 (pH 2)	20.5–32.9 (pH 2) 3.5–5.3 (pH 7)	14.5–18.5 (pH 2) 18.1 (pH 7)	[56,63,71,73,130,151,156]
Glucose adsorption capacity (mmol/g)		2.3 (50 MM)	2.0 (50 MM)	[63,71,154]
		2.1–7.3 (100 MM)	4–5 (100 MM)	
		15.7 (200 MM)	8.0 (200 MM)	
Sodium cholate adsorption capacity (mg/g)	60.6–67.5	3.2	10–37	[63,73,130,142,151]
Cation exchange capacity (mmol/g)		0.51	0.122–0.132	[63,73,154]
Nitrite ion adsorption capacity (µmol/g)			37 (pH 2) 15 (pH 7)	[73]
Viscosity (mPa·s)	1.27–1.33			[62,112]
Viscosity (CentiPoise)	570			[157]

#### 2.2.2. Physico-Chemical Properties of Fibre-Associated Phenolic Compounds

Among WB compounds there are different types of antioxidants, such as iron, zinc, copper, and selenium, which act as cofactors of antioxidant enzymes. The direct antioxidants are polyphenols (including lignans, anthocyanins and alkylresorcinols), lignins, vitamin E, and carotenoids [158]. Lignin polymers are not represented in high quantities in WB, and they are probably not responsible for the antioxidant activity because they are bound to biomass. Polyphenols have antioxidant activity, which is interesting for health effects as protection against oxidative stress. Mechanisms of polyphenol antioxidant activities include suppression of reactive oxygen species (ROS) formation by inhibition of enzymes, scavenging ROS, or upregulation of antioxidant defences [159]. 

Total phenolic content (TPC) analysis is mainly measured with colourimetric and chromatographic methods. Phenolic compounds are important oxygen scavengers. The Folin–Ciocalteu colourimetric assay measures the reduction of phosphotungstic and phosphomolybdic acids content in molybdenum and tungsten oxide by the sample’s phenols [160], TPC being quantified against a gallic acid standard curve. Results represent monophenols but also include proteins with tyrosine residue [161] or other substances (reducing sugars or ascorbic acid) with reducing properties for the Folin–Ciocalteu reagent, making the test rather nonspecific [162]. Thus, total reducing activity would be more accurate than TPC. That is why polyphenols may be analysed more precisely by HPLC coupled with diode array-type detector (DAD-PDA) and/or tandem mass spectrometry (LC-MS/MS) because it can separate the compounds and quantify them one by one. LC-MS/MS enables both noise reduction and sensitivity improvements by exploiting the multiple reaction monitoring scan mode (MRM) [163]. 

The extraction of polyphenols is based on their polarity, acidity, volatility, and size. Acidification with high temperature serves to liberate linked polyphenols. Phenol flavours are volatile, so they can be extracted by solid phase microextraction (SPME). Flavonoids, stilbenes and chalcones, non-volatiles compounds, may still be analysed by SPME after a silylation with bis(trimethylsilyl)trifluoroacetamide (BSFTA) [160]. Liquid–liquid and solid–liquid extractions followed by concentration and purification stages are the most widely used methods [163]. Ethanol, methanol, and water are the better solvents to extract phenolic compounds and flavonoids [164]. In wheat bran, most of the polyphenols are bound and account for about 75% of the TPC [44,89] (Table 3). For bound phenolic compounds (BPC), acidic, alkaline or mixed hydrolyses are employed, and then BPC are extracted in organic solvent (diethyl ether and ethyl acetate). Alkaline hydrolysis treatment with NaOH on soluble extract allows separation of the soluble conjugated FA [89]. Sometimes methanol extraction is coupled with HCl extraction to favour anthocyanins, which are more extractable when the pH is under 2. Aqueous acetone is preferred for flavonoid extraction because it breaks hydrogen bonds [97]. Before or during hydrolysis, n-hexane treatment is applied in order to eliminate lipids [95]. Freezing at −20 °C improves polyphenol purification by precipitating proteins. Ascorbic acid and EDTA can be used to prevent phenolic acid degradation [165]. 

Lignin detection can be done using Klason lignin, which is the official method, or thioacidolysis, which disrupts non-condensed intermonomer linkages, and monolignols are analysed by GC-MS after extraction [74]. Elemental analyses of lignin are the determination of residual carbon content, ash, and molar mass (size-exclusion chromatography), but it may be interesting to analyse lignin by nuclear magnetic resonance (NMR) spectroscopy because of lignin’s complex molecular structures and the interest in studying the molecular bonds of the polymers. Physico-chemical properties are determined by thermogravimetric analysis (TGA) and differential scanning calorimetry (DSC) to identify functional groups and density [166].

##### Antioxidant Activities

Phytochemicals, such as phenolic compounds, are known to prevent oxidative stress by maintaining a balance between oxidants and antioxidants. Determination of the antioxidant activity of polyphenols, as well as their ability to eliminate free radicals using antioxidant assays, is widely used for plant extract characterisation. These methods allow us to know if the matrices contain bioactive compounds with antioxidant properties that promote health by protecting against oxidation. Free-radical-scavenging antioxidant assays are: 2,2-Dipheny-1-pucrylhydrazyl (DPPH) radical scavenging capacity and 2,2′-Azino-bis (3-ethylbenzothiazoline-6-sulfonic acid) (ABTS). DPPH is the usual method [164]. There are two possible mechanisms of action: hydrogen atom transfer (HAT) or single electron transfer (SET), depending on if the radical is neutralised by accepting either a hydrogen atom or an electron from antioxidant species [162]. The reaction time of 30 min allows DPPH to react even with weak antioxidants. The antioxidant capacity of the sample is expressed as the half of maximal inhibitory concentration (IC_50_), which is the concentration of a sample antioxidant molecule needed to reduce half of the DPPH. Therefore, the lower the IC_50_ is, the higher the antioxidant activity of the sample. EC_50_ is another way to express antioxidant activity; it refers to the steady-state kinetics and it represents the concentration of an antioxidant needed to decrease the initial absorbance of DPPH radicals by 50%. Antioxidants with fast kinetics have the same value for EC_50_ and IC_50_, while different values for EC_50_ and IC_50_ correspond to slow-kinetics antioxidants [167]. Some researchers express the antioxidant capacity as mass of standard equivalent per mass of dry weight. The main antioxidant standards used as positive controls are Trolox, ascorbic acid (Vitamin C), or gallic acid [168]. The higher the standard equivalent concentration, the more the antioxidant capacity is dependent on the extraction method; wheat bran exhibits an average antioxidant activity value of about 7.6 μmol trolox equivalent (TE)/g [169,170]. The highest antioxidant activity is obtained for 80% ethanol extracts which, interestingly, contained the highest quantity of phenols as bioactive compounds. DPPH radical scavenging ability was demonstrated to be dependent on the studied varieties [171]. The variation in antioxidant activity between wheat varieties may be related to differences in composition of the phenolic compounds. Obviously, the antioxidant activity depends on the extraction conditions, extraction solvents, wheat bran granulometries, and wheat bran composition (i.e., histological composition). 

DPPH and ABTS are strongly correlated, and ABTS shows higher results than DPPH [172]. ABTS is applicable for hydrophilic and lipophilic compounds. The antioxidant sample will give an electron to reduce ABTS^•+^ into ABTS (stable form) for the SET mechanism, which is mainly involved in the elimination of ABTS free radical [73,168]. Absorbance is read at 725/734 nm and expressed in a standard equivalent quantity, which can be Trolox, ascorbic acid, gallic acid, butylated hydroxyanisole (BHA), or butylated hydroxytoluene (BHT) [173]. The average ABTS radical scavenging activity is about 142 μmol TE/g for wheat bran extracts. The absolute methanol and 80% ethanol extracts show the highest antioxidant activity [170]. With ABTS and DPPH methods, numerous expressions of results are used, which makes comparison between the research findings very difficult. Whatever the expression used, the ABTS radical scavenging activity varies according to the fraction of bran used (particle-size distribution), the fibres analysed, the nature of the polyphenols (i.e., free or bound), the extraction solution, and of course the genotype and environment factor (Table 5).

Another type of antioxidant analysis can be applied to wheat extracts; it is a reducing potential antioxidant assay based on metal reduction of iron or copper. The one utilised here is the ferric ion reducing antioxidant power (FRAP), because wheat FA and *p*-coumaric acid can chelate protonated metal ions such as Cu(II) or Fe(II) [174].

The FRAP assay measures the reduction of ferric ion Fe^3+^ (colourless) in Fe^2+^ (blue) under acidic conditions. The impact of the antioxidant sample is uncertain because it is still unclear whether the antioxidant will scavenge free radicals or chelate with metal ions. As for the ABTS assay, FRAP activity is expressed in standard equivalent quantity. Standards can be Trolox, gallic acid, ascorbic acid, quercetin, or α-tocopherol [162]. The antioxidant power of bran extracts evaluated by the FRAP assay ranges between 8.9 and 54 μmol of FeSO_4_/g defatted bran (Table 5). Variations are often due to wheat bran fractionation, particle size distribution, wheat bran treatment, the composition of extracts (free and or bound polyphenols), genotype factor, and extraction solvent [175,176,177]. All these parameters affecting FRAP values reflect variations in the composition of polyphenol molecules. 

Total antioxidant activity can also be determined by total oxyradical scavenging capacity analysis (TOSC). Oxygen radical absorbance capacity (ORAC) is a HAT assay widely used in food antioxidant determination. It has been adapted to detect both hydrophilic and lipophilic compounds, but it has some drawbacks which make it not reflective of the antioxidant capacity. Adom and Liu (2002) used 2,2′-azobis-amidinopropane (ABAP) to oxidise α-keto-γ-methiobutyric acid (KMBA) to form ethylene gas [89]. The degree of inhibition of ethylene gas is measured by Gas Chromatographic HeadSpace analysis (HS-GC). Only the gas above the sample is introduced in the GC column [178]. The antioxidant capacity is expressed in IC_50_ vitamin C equivalent. 

**Table 5 foods-12-02693-t005:** Antioxidant values in literature for DPPH, ABTS and FRAP tests on wheat bran.

Antioxidant Capacity Value	Wheat Bran Sample	References
DPPH		
59.42%	SDF	[71]
2.91%	IDF residue	
84.11%	IDF bound polyphenols	
64.7%	Wheat IDF	[73]
3.6 mm TEAC/g	Wheat bran (free)	[179]
17.7 mm TEAC/g	Wheat bran (bound)	
20.0 mm TEAC/g	Wheat bran (total)	
6.42 EC50 mg/mL	Wheat bran	[167]
7.02 IC50 mg/mL	Wheat bran	
5.2% (discolouration)	Wheat bran (free)	[96]
9.7%	Wheat bran (acid hydrolysis)	
15.0%	Wheat bran (alkaline hydrolysis)	
4.2–4.7 µmol TE/g DW	Wheat bran (free)	[44]
7.5–8.3 µmol TE/g DW	Wheat bran (bound)	
11.7–13.0 µmol TE/g DW	Wheat bran (total)	
14.5% (inhibition)	Wheat bran coarse (soluble)	[175]
15.9%	Wheat bran medium (soluble)	
13.7%	Wheat bran fine (soluble)	
41.3%	Wheat bran coarse (bound)	
43.0%	Wheat bran medium (bound)	
32.6%	Wheat bran fine (bound)	
ABTS		
88.42%	Wheat IDF	[73]
10.2 mm TEAC/g	Wheat bran (free)	[179]
40.5 mm TEAC/g	Wheat bran (bound)	
50.7 mm TEAC/g	Wheat bran (total)	
6.6 IC50 mg/mL	Wheat bran	[167]
2.9 µM trolox equivalent	Wheat bran (free)	[96]
6.0 µM trolox equivalent	Wheat bran (acid hydrolysis)	
9.1 µM trolox equivalent	Wheat bran alkaline hydrolysis	
FRAP		
53.04 µmol FeSO_4_/g	Wheat bran (coarse)	[175]
40.84 µmol FeSO_4_/g	Wheat bran (fine)	
23.8 µmol FeSO_4_/g of defatted bran	Wheat bran coarse (soluble)	
8.9 µmol FeSO_4_/g of defatted bran	Wheat bran medium (soluble)	
12.3 µmol FeSO_4_/g of defatted bran	Wheat bran fine (soluble)	
229.2 µmol FeSO_4_/g of defatted bran	Wheat bran coarse (bound)	
28.3 µmol FeSO_4_/g of defatted bran	Wheat bran medium (bound)	
28.6 µmol FeSO_4_/g of defatted bran	Wheat bran fine (bound)	
53 nmol TE/g grain	Wheat bran	[167]
58.4 µmol/g	Soluble AX from wheat bran	[180]
21.42 µmol TE/g	SDF	[139]
0.36 mmol/L	SDF	[71]
0.09 mmol/L	IDF residue	
1.59 mmol/L	IDF bound polyphenols	
11.0 mm TEAC/g	Wheat bran (free)	[179]
34.5 mm TEAC/g	Wheat bran (bound)	
48.9 mm TEAC/g	Wheat bran (total)	

## 3. Heath Benefits of Wheat Bran

### 3.1. General Effects Linked with WB Functional Properties

Wheat bran has physiological effects on humans which can be grouped into two major categories: Effects directly linked to physico-chemical properties of wheat bran, and effects linked to wheat bran fermentation by colonic microbiota. It should be noted that physico-chemical properties of WB fibre influence WB fermentation, but their consequences can be observed in the upper parts of the digestive tract. We will review these effects focalising on AX, β-glucans and polyphenols, as they are considered to be the main bioactive compounds in wheat bran. The mechanisms involved in producing health benefits are not entirely understood. Several physico-chemical mechanisms could be involved, but they are dependent on various factors such as composition, molecular properties, functional activities (Section 2.2) and physical forms of dietary fibre. It is important to understand these mechanisms controlling health effects by studying mechanisms by which macronutrients, mainly starch and lipid, are released and digested in the presence of different kind of dietary fibre.

Physiological properties of DF are linked with their physico-chemical characteristics (particle size, branching degree, monosaccharide and linkage composition, Mw, WHC, WSC, WBC, solubility, OBC, viscosity, and gel-forming capacity). According to literature, these molecular properties will help to reduce blood cholesterol and glucose levels, limit the risk of chronic disorders (cardiovascular diseases, diabetes, obesity, and some forms of cancer) and lead to laxative effects [108] (Figure 4). The hydration properties are related to fibre fermentation in the digestive tract. Low WBC observed for highly compacted DF could limit fermentation due to the compactness of polysaccharides within the granules [181]. DF have good WSC, they absorb water and thus enhance satiety and reduce food intake. But this property can also delay nutrient intake. Since DF have lubricating effects thanks to the WHC property, they can promote peristalsis and intestinal motility. The water absorption by DF leads to an increase in stool volume and promotes defecation. Cellulose and water-insoluble AX are major contributors to laxative effects and faecal bulking. IDF increase faecal bulk and slow intestinal transit due to their porosity and low density. SDF consumption is associated with a decrease in postprandial glycaemic and lipidemic responses thanks to their ability to delay gastric emptying, and subsequently cholesterol, glucose and nutrient absorption. Chen et al., observed a better CAC in SDF than IDF [71]. Indeed, in vivo administration of SDF increased by 35 to 65% the excretion of bile acid and salt into the faeces, leading to total plasma cholesterol (TC) and LDL-cholesterol reduction, but did not affect HDL-cholesterol. SDF also reduces cholesterol by reducing glucose absorption, which will reduce insulin production [182]. Insulin favours lipogenesis and inhibits lipolysis, so that reduced insulin secretion leads to the reduction of lipogenesis. Insulin also favours leptin liberation by adipocytes, so we could hypothesise that insulin inhibition by SDF would reduce satiety, but the opposite effect is observed. The satietogenic effect is due to the expansion of food volume by fibre hydration and the production of short-chain fatty acids (SCFAs) which can promote the release of satiety hormones [119]. 

AX are fermented by the colonic microbiota; the unsubstituted xylose regions are fermented preferentially, followed by the branched oligosaccharides [119]. AX biological effects are closely associated with their molecular size or the linkage distribution. According to the literature, AX enhances immunomodulatory, antitumor and antioxidant activities, and postprandial metabolic capacity on glucose response and lipid metabolism. Indeed, it reduces the postprandial serum glucose, insulin and plasma total ghrelin, which is an appetite stimulator [183]. 

The gelling properties of β-glucans are linked with their capacity to increase the viscosity of digesta and to delay the absorption of nutrients in the small intestine. Viscous DF has good WHC, which blocks the flow of carbohydrates in the digestive tract and is responsible for slower gastric emptying, resulting in improving satiety. High Mw β-glucans are more efficient because they have higher viscosity [184]. Further, β-glucans are resistant to gastric and pancreatic enzyme digestion; they are fermented by colonic bacteria in the cecum and colon to produce SCFA.

### 3.2. Effects on Glucose Metabolism

For the glycaemic answer, Ou et al. (2001) reported three ways of lowering postprandial serum glucose levels: (i) the viscosity in the small intestine hinders glucose diffusion in the plasma, (ii) binding of glucose, which limits glucose adsorption, and (iii) inhibition of α-amylase activity, postponing the release of glucose from starch by encapsulating starch and the enzyme [185]. If the α-amylase activity is inhibited, it will reduce the carbohydrate conversion to glucose [63]. SDF and IDF both can lower postprandial serum glucose levels by encapsulating α-amylase and starch to delay substrate accessibility or by directly inhibiting α-amylase activity [139].

A reduction of blood glucose levels in type-2 diabetes (T2D) patients after β-glucan consumption in cereals has been reported with a reduction of glycosylated hemoglobin (HbA1c) [186,187]. The viscosity is considered responsible for hindering glucose and lipid absorption, but other mechanisms are involved. Aoe et al., in their 2020 preclinical study, suggest that low Mw β-glucan improves glucose and lipid metabolism through prebiotic effects [49]. They suppose that glucose concentration might be lowered through GLP-1 action enhanced by SCFA. Indeed, SCFA stimulate the release of GLP-1 incretin hormones via the G protein-coupled receptor 43 in endocrine cells. GLP-1 promotes insulin secretion [188,189]. In a clinical study, Pino et al. showed that β-glucan consumption leads to reduced GLP-1 and insulin production in T2D patients [187]. Bays et al. (2011) have shown increasing insulin resistance in populations at risk for T2D [190]. Serum insulin concentration is reduced with high Mw β-glucan and also the expression of hepatic SREBP-1c (sterol regulatory element binding protein-1c), which is a transcription factor controlling the process of de novo fatty acid synthesis [191]. These effects are due to the suppression of carbohydrate digestion and absorption.

AX consumption in overweight subjects reduced plasma total ghrelin significantly without modification of plasma-acylated ghrelin [192]. Acylated ghrelin (AG) is associated with higher glucose levels and inhibition of insulin secretion, whereas the unacylated form (UAG) antagonises these effects. UAG and AG have cardiovascular effects, cell proliferation modulation and influence adipogenesis. Ghrelin favours the development of obesity and obesity-associated type 2 diabetes [193,194]. This effect is associated with the post-prandial response of insulin and serum glucose lowering. Mio et al. (2022) observed in their preclinical study an increase in GLP-1 secretion enhanced by SCFA increase linked with barley AX consumption by mice [189]. Lafond et al. (2015) described an increase in GLP-1 secretion in overweight women who ate a high-fibre breakfast. This occurs in the distal small intestine with proteins and carbohydrate digestion, which are both stimulators of GLP-1., in the distal small intestine [195]. Additionally, in the later interval between meals, the GLP-1 secretion is enhanced by the stimulation of L-cells by SCFA released during fermentation. Peptide YY is located with GLP-1 on L-cells (entero-endocrines cells) in the distal small intestine [196]; it is a satiety hormone which increases with a high fibre diet. Increasing of PYY and GLP-1 improves insulin sensitivity, as we saw for β-glucans.

### 3.3. Effects on Lipid Metabolism

Gulati et al., in 2017, affirmed that the consumption of SDF leads to beneficial effects on human lipid parameters, specifically total cholesterol and low-density lipoprotein cholesterol (LDL-C) [197]. These positive effects appear to be related to β-glucan. The health benefits of β-glucans were advocated by the U.S. Food and Drug Administration (FDA) in 1997 to reduce the risk of heart disease and to control blood cholesterol levels [198]. They authorised products with at least 3 g of β-glucans to be registered as “cholesterol-reducing products” [199]. We focus on oat and barley β-glucan because most of the studies on β-glucan use these cereals. However, β-glucan from wheat has a similar structure but may differ in quantity and Mw. Indeed, wheat β-glucan intrinsic viscosity is better than that of oats (3.5 and 1.3 dL/g, respectively), while barley shows the highest viscosity, at 4.1 dL/g [49]. 

Keenan et al. and Wolever et al. both observed a reduction of LDL-cholesterol linked to oat and barley β-glucan in their clinical studies [200,201] (Table 6). In general, we can observe a reduction of total and LDL-cholesterol without modifying the HDL-levels and triglyceride concentrations. Wolever et al. showed that oat β-glucan was more efficient for Mw above 210,000 g/mol [201]. Thandapilly et al. observed an elevation of total faecal SCFA, which promotes hindgut fermentation due to barley β-glucan. This can attenuate cholesterol levels [202]; the authors also noticed upper levels of the bile acid lithocholic acid excretion for 3 g per day high-Mw β-glucan. Bile acids derive from cholesterol; they are necessary in the small intestine for the digestion of lipids [151]. 

AX counterbalance high-fat diet-induced hypercholesterolemia in mice because they decrease cholesterol absorption, increase faecal excretion of cholesterol bile acids, and lead to higher propionate secretion [203]. AX downregulate genes involved in adipocyte differentiation, fatty-acid uptake, fatty acid oxidation, lipolysis, and decrease fatty acid synthesis. AX limits cholesterol and fatty acid absorption thanks to its fat-binding capacity, which will lead to an anti-obesity effect [203]. AX chain reticulation by oxidative coupling with FA in the presence of a strong oxidant and ROS producers leads to gel formation. These gels have high WRC (100 g of water/g of dry gel) [54]. Garcia et al. (2007) have shown triglyceride level reduction on humans after AX consumption [192]. On overweight patients, lipid concentrations are not affected [204]. Low molecular weight AX (AXOS) do not provoke effects on lipid metabolism according to clinical studies [205,206,207]. As β-glucans, dietary AX reduce the plasma total cholesterol and LDL-cholesterol concentrations by inhibiting the cholesterol absorption and promoting the excretion of bile acids [192].

### 3.4. Effects on Microbiota 

Even though fibre is resistant to hydrolytic digestion by human enzymes, it is fermentable, making wheat bran an important nutrient source for intestinal microbiota. Gut microbiota plays a crucial role in human health: improvement of intestinal barrier function, regulation of host metabolism, intestinal homeostasis, and a strengthened immune system. Colorectal cancer, CVD, type-2 diabetes, obesity, and inflammatory bowel disease are associated with intestinal flora dysbiosis [119]. The flora composition can be affected by environment, diet, or drugs. In general, easily fermentable compounds are fermented in the proximal colon where they promote SCFA production and increase lactic acid bacteria populations [208]. Acetate, propionate and butyrate are the main SCFA; they occur in molecular ratios of 60:20:20 in the colon [209]. These molecules regulate inflammatory response (depending on their concentration) and promote the regeneration of intestinal cells [57,210]. It should be noted that depending on the fibre type (pectin, AX, β-glucan, or resistant starch), we can observe differences in faecal microbiota composition and SCFA production [211]. Proliferation of *Bifidobacterium* and *Lactobacillus* leads to a prebiotic effect [212]. Conversely, fermentation of slow-fermentable compounds occurs beyond the proximal colon where protein fermentation is prevalent [213]. Also, SCFA promote Glucagon-Like Peptide 1 (GLP-1) and YY peptide (PYY) secretion, leading to insulin secretion and increasing satiety, which can reduce obesity [214]. Wheat bran, AX and AXOS effects on microbiota have been well reviewed by Jefferson and Adolphus [215]. Daily fibre intake leads to increased bacterial population and diversity. Wheat bran, because of its complex structure, presented a more interesting effect to promote microbial diversity. The effects on microbiota depend on fibre intake (isolated or not), starting microbiota of individuals and consumption duration [215].

Literature reports effects of β-glucan on microbiota composition, but effects are not obvious. β-glucan fermentation by the gut microbiota depends on its physiochemical structure [216]. On preclinical studies in mice, β-glucan can have effects on the proliferation of *Lactobacillus*, *Bifidobacterium* and *Bacteroides*, depending on grain variety (barley, oat) [217,218]. The increase in beneficial bacteria leads to increased SCFA. A high Mw β-glucan diet increases fibre fermentability more than low Mw on mildly hypocholesterolemic subjects with higher SCFA production [203]. SCFA decrease serum cholesterol and fasting blood glucose levels, and boost leptin concentration [210]. The link between leptin and SCFA is still unclear [219]. There is a relation between SCFA climbing and leptin increasing in in vitro studies, but not in in vivo studies in which the increase in SCFA is correlated with a decrease in leptin [219]. Clinical trials on β-glucan microbiota modulation do not show the same tendency. A recent clinical trial reports a decrease in butyrate-producing bacteria and *Lactobacillus* spp. after oat β-glucan consumption by T2D patients [187]. This is also the case for barley β-glucan consumption. A decrease in gastrointestinal microbial diversity has been observed on volunteers with a high risk of metabolic syndrome development [220].

Wheat AX is degraded by colonic bacterial enzymes (xylanases, arabinofuranosidases) to produce short-chain AX called arabinoxylooligosaccharides (AXOS), which have lower viscosity due to their lower Mw. Highly feruloylated AXOS are less fermented because associated FA can inhibit fermentation. Bound FA can sterically hinder microbiotic enzyme activities, and free FA can exert antimicrobial effects [221]. AXOS have beneficial effects on prebiotic, immunomodulatory, and fermentation activities. In general, AXOS consumption increases butyrate-producing bacteria. Kjølbæk et al. reported that AXOS intake reduces *Rikenellaceae* and *Porphyromonadaceae* bacteria species which are associated with inflammatory processes, and it increases bifidobacteria and clostridia species which are butyrate producers [208]. For clostridia species, this observation is not confirmed because there is usually no change reported [206,222,223,224,225]. However, the effect of AXOS on bifidobacteria levels has been confirmed after 2, 3 and 4 weeks of AXOS intake in other clinical studies [206,222]. Bifidobacteria increase linked with AXOS intake leads to elevated SCFA secretion, in particular butyrate production [222]. Butyrate has been reported to have chemopreventive effects thanks to anti-inflammatory and immunomodulatory properties. It is an energy source for colonocytes and is essential for colonic homeostasis; it decreases oxidative stress and reinforces the colonic defence barrier via mucins and antimicrobial peptide secretion, for example [226,227]. Damen et al. (2012) observed increased faecal total SCFA concentrations with a butyrate upsurge of 70% in healthy volunteers [223]. According to Van Craeyveld et al. (2008), the bifidogenic potency is influenced by AXOS DP [209]. Low DP is more efficient on bacteria proliferation. The fermentation of AX is also influenced by the A/X ratio and the presence of phenolic acids; low A/X ratio and low di-substituted backbone are more efficiently and more quickly fermented, respectively [227]. AX Mw also seems to affect fermentation; Salden et al. (2018) observed no change in microbial profiles in faecal samples after 6 weeks AX consumption and a reduction of faecal SCFA concentration [205]. 

### 3.5. Immunomodulatory Activity

Regarding the immune system, Estrada et al. (1997) showed β-glucan immunomodulatory effects in vitro by the increase in IL-1α cytokine production by murine macrophages in the presence of oat β-glucan [228]. Cytokines are responsible for immune and inflammation response modulation. Estrada et al. also observed a higher secretion of IL-2, INFγ and IL-4 in spleen cells. IL-2 and INFγ have a role in cell-mediated immunity via cytotoxic and helper T-lymphocytes. In contrast, IL-4 is involved in the humoral immune response, which involves the production of antibodies by B-lymphocytes against pathogenic antigens [229]. The immunologic effect of β-glucan is attributed to its structure, water-solubility, viscosity, composition, and Mw [230,231]. Clinical trials on humans to study immune and antiproliferative effects of β-glucan are mostly made with yeast β-glucan, which are different to cereal glucans (1,3- 1,6- branched) [229,232,233].

AX fermentation effects concerning stimulation of innate responses have been reported. Despite the effect of SCFA synthesis on immunomodulation, some in vitro and preclinical studies show that AX can directly or indirectly activate immune cell proliferation that will produce cytokines [76,234,235]. In humans, AX consumption for 6 weeks decreased the pro-inflammatory cytokine TNFα in stimulated Peripheral Blood Mononuclear Cells (PBMC) but did not change other cytokines (IL-2, IL-10, INFγ, IL-12p40); this means that AX seems to have anti-inflammatory effects [205].

In healthy volunteers, a reduction of urinary phenol and *p*-cresol excretion after AXOS consumption, compared to the control, reached more than 50% for *p*-cresol after 10 g AXOS/day and 37% after 10 g of WB consumption per day [206,207,222]. *p*-cresol corresponds to 90% of the urinary phenolic compounds and is an indicator of protein fermentation, which is detrimental for host health and is considered to be the prototype of protein-bound uremic toxins in chronic kidney disease [207]. Indeed, proteolytic fermentation products are toxic phenolic compounds, which are bacterial metabolites from tyrosine not produced by human enzymes. They are detoxified by the liver and excreted in urine [206]. This detoxifying activity has been characterised in a preclinical study which demonstrated a modulation of the Nrf2/antioxidant response element pathway which induces detoxifying/antioxidant activity [236].

In addition to favourably modulating intestinal fermentation, AXOS plays a role on human gastrointestinal (GI) proper modulation. Damen et al. (2012) showed that the consumption of bread with AXOS enhances stool frequency [223], while high Mw AX do not have any effect on GI modulation [205]. Both AX and AXOS are well tolerated by humans [205,222].

### 3.6. Antiproliferative Activity on Cancer 

The antitumor activity of β-glucan depends on its Mw and degree of branching [230,231]. Shah et al. (2017) classified β-glucan as a good antioxidant and antiproliferative agent against cancer [237]. The immune and antiproliferative effects of cereal β-glucans are not elucidated yet but could be linked to gut microbiota interactions.

There are no clinical trials confirming the antiproliferative effects of AX on humans, but some in vitro studies reveal a potential inhibitory effect on cancer cells, as shown by Paesani et al. in 2021 on HCT-116 colon cancer cells [238]. They showed decreasing viability of cancer cells with AX without impacting non-cancer cells’ viability. Non-cancer cells were immune cells (macrophages and spleen cells) and their viability increased a little, suggesting an immunomodulatory effect and a selective effect of AX on cancer cells. They observed a better effect in AX from hard wheat than soft wheat, and they supposed it was becausee of the higher content of α1 → 2 and α1 → 3 bonds and/or greater amounts of xylose disubstituted with arabinose residues in hard wheat AX. Femia et al. (2010) showed a reduction of preneoplastic lesions in rats treated with a colon carcinogen and fed with AXOS compared to the ones fed with a control diet [239]. AXOS seems to have a better efficiency on cancer cell inhibition than AX, as investigated by Mendez-Encinas et al. in 2021 [227]. They worked on maize AX, which have the same structure as wheat AX but have more FA. The investigators observed a dose-dependent inverse effect on Caco-2 colon cancer cells but no inhibition of normal human colon cancer cells (CCD 841 CoN). The antiproliferative activity is related to the AX concentration, the SCFA production, and the FA content. Soluble AX decreases H_2_O_2_ damage by 64% on HT29 colon cancer cells. They also increase SCFA, in which acetic acid was the most represented in Glei et al. and Mendez-Encinas et al. articles [227,240]. 

### 3.7. Phenolic Compounds Effects on Health

The effects of wheat bran on health can also be linked to their phenolic compounds, which are well known for their antioxidant properties but also have effects on glucose metabolism, lipogenesis, and the immune system. Usually, polyphenols are considered capable of preventing cancer development, CVD, and neurodegenerative diseases based on in vitro studies, which most of the time isolate polyphenol components of the food matrix [163,241]. In 2010, Vauzour et al. reported that there are not sufficient in vivo long-term studies in humans to prove these health effects [242].

Phenolic acids have antioxidant activity. In wheat bran, the antioxidant capacity is closely connected with the level of FA. It leads to cellular defence and contributes to the protection of cells from oxidative stress damage. The high antioxidant capacity of bound polyphenols can be responsible of IDF antioxidant capacity since we know that they are mostly linked to IDF. However, higher DPPH scavenging activity have been reported in SDF in comparison to IDF [71,173]. This is due to SDF higher content of xylose which have hydroxyl groups that provide radical scavenging activity [71]. 

Free, and some conjugated, FA can be absorbed in the human small and large intestine conversely to bound FA which are linked to NSP and are not easily bioavailable [99]. DF can lower polyphenols released from the food matrix in the upper digestive tract, thereby increasing the polyphenol amount that reaches the lower digestive tract where they will beneficially modulate microbiota and potentially help to prevent colon cancer, according to in vitro studies [243]. During colonic fermentation, the bioaccessibility of SDF-bound polyphenols is increased by 7.4 times compared to gastrointestinal stage according to in vitro studies, and this increased antioxidant activity [244]. Free phenolic acids can be absorbed via paracellular and active transport mediated by monocarboxylic acid transporters in the gastrointestinal mucosa. The affinity of the different phenolic compounds with this transporter varies the absorption efficiency [245]. 

DF attenuates post-prandial blood glucose levels by different mechanisms; we saw previously the viscosity parameter of SDF (AX and β-glucans) acting as a physical barrier, which retards the absorption of glucose and restricts the enzymes’ access to starch. DF can inhibit starch digestion by encapsulating it, limiting gelatinisation, and maintaining crystallinity and short-range ordered structure [246]. However, polyphenols also play a role in the reduction of starch digestion by improving insulin sensitivity and inhibiting α-amylase. Chen et al. (2021) showed a higher α-glucosidase inhibitory activity on bound polyphenols, which contributes to the hypoglycaemic power of IDF [13,71,79,247]. 

Overall, polyphenols reduce lipogenesis, increase lipolysis, and inhibit adipogenesis in vivo in preclinical studies. Polyphenol anti-obesity effects have been investigated and confirmed in humans [248]. This will limit lipid accumulation and reduce CVD. Atherosclerosis is an inflammatory disease caused by fatty deposits in the arteries. These will create atherosclerotic plaques and can cause CVD. The disease is also associated with LDL accumulation in the circulation and production of oxidised LDL (oxLDL). Polyphenols limit atherosclerosis progression because they modulate vascular and endothelial functions by improving the lipid profile by the reduction of LDL-cholesterol [249]. 

Phenolic compounds enhance the immune system by alleviating proinflammatory cytokines, chemokines, and angiogenic factors. They modulate enzymes associated with B-cell activation (antibody productive lymphocytes) and T-cell proliferation (destructive pathogen cells lymphocytes) [249]. Isolated phenolic compounds such as gallic acid, chlorogenic acid, and kaempferol (100 µmol/L) inhibit LPS-induced infiltrated macrophages by the reduction of iNOS and COX-2 expression [250]. Phenolic compounds also play a role in detoxification; FA increases activities of glutathione S-transferase and quinone reductase in the liver and colon (100 mg FA/kg body weight). This suggests that the detoxifying activity of these enzymes is related to the effect of FA on colon carcinogenesis [251].

In summary, β-glucan, AX and associated polyphenols seems to have synergic effects on biological parameters. Health effects of β-glucan validated by clinical trials are reduction of blood total and LDL-cholesterol, glycemia, satiety, improving insulin sensitivity, and increasing SCFA production. Cereals β-glucan effects on microbiota, the immune system and antiproliferation are not clear and need to be further investigated. In addition, validated effects are on oat and barley β-glucan rather than wheat. For AX, we can find effects on glucose and cholesterol metabolism, the immune system, and a prebiotic effect. Biological effects are more characterised in particular for the prebiotic effect, which was not well identified for β-glucan. 

**Table 6 foods-12-02693-t006:** Clinical studies observations for β-glucan and AX effects on health. “↗” is for increase, “↘” for decrease and “=” for “No change”.

Type of Fibre, Quantity	Clinical Study Duration	Volunteers (Disease, Number, Sex)	Results Linked with Fibre Consumption	References
High and low Mw concentrated barley β-glucan extract 3 g and 5 g doses	10 weeks	Hypercholesterolaemia 155 M/F	↘ LDL-Cholesterol ↘ Total cholesterol = HDL Cholesterol levels	[201]
High and low Mw concentrated barley β-glucan 6 g/day	6 weeks	Hypercholesterolemia 90 M/F	Low Mw group ↘ Ratio TC/HDL ↗ Body weight ↘ Hunger High Mw group ↗ Ratio TC/HDL ↘ Body weight ↘ Hunger = Blood pressure, glucose, insulin, gastrointestinal symptoms	[252]
5 groups: A: 3 g/day wheat fibre B: 3 g/day oat high Mw β-glucan C: 4 g/day oat medium Mw β-glucan D: 3 g/day oat medium Mw β-glucan E: 4 g/day oat low Mw β-glucan	4 weeks	5 mmol/L > LDL-C > 3 mmol/L 345 M/F	LDL-C in groups B, C and D < group A Non-significant for sex, age, baseline LDL-C	[202]
3 g barley high Mw β-glucan 5 g barley low Mw β-glucan 3 g barley Low Mw β-glucan	5 weeks	Mildly hypercholesterolemic subjects 30	↗ Lithocholic acid excretion ↗ Fermentability of fibre ↗ Total faecal SCFA in group 3 g/d high Mw	[203]
Control 3 g/d reduced viscosity barley β-glucan 6 g/d reduced viscosity barley β-glucan	12 weeks	Healthy subjects 50	↗ Insulin sensitivity for 6 g/d = Body weight	[191]
Control: microcrystalline cellulose Oat β-glucan (5 g/day)	12 weeks	Type-2 diabetes patients 37	↘ HbA1c ↘ Insulin and GLP-1 ↘ C-peptide ↘ HOMA ↘ *Lactobacillus* spp. and Butyrate-producing bacteria	[187]
Control group: Placebo 15 g AX	6 weeks	Impaired glucose tolerance, insulin resistant, slightly elevated LDL and low HDL cholesterol concentration. Body mass index (BMI): 30.1 kg/m^2^ 7 women 4 males	↘ Postprandial response in serum glucose, insulin, triglycerides, plasma total ghrelin. = Plasma acylated ghrelin	[193]
7.5 g/day AX (*n* = 16) 15 g/d AX (*n* = 17) 15 g/d placebo (control *n* = 14)	6 weeks	Overweight 47	= Gastrointestinal permeability and tight junction ↘ Faecal microbiota diversity ↗ Faecal pH ↗ Faecal concentration total SCFA ↘ TNFα by simulated PBMCs	[205]
10 g placebo (control) 10 g wheat AXOS	3 weeks	Healthy subjects 20	= Blood lipid = Gastrointestinal symptoms ↗ Flatulence ↗ Bifidobacteria ↘ Urinary *p*-cresol (after 2 weeks)	[206]
0, 3 and 10 g/day wheat bran extract enriched in AXOS	3 weeks	Healthy volunteers 63	↘ Urinary *p*-cresol (10 g/d) ↗ Faecal bifidobacteria (10 g/d) ↗ Faecal SCFA ↗ Flatulence frequency distress severity (10 g/d) ↘ Faecal pH ↘ Constipation occurrence frequency	[207]
10.4 g/d AXOS from wheat bran extracts and 3.6 g/d polyunsaturated fatty acids	12 weeks	Overweight individuals with indices of metabolic syndrome 55	↗ Bifidobacteria ↗ Butyrate producing bacteria	[208]
Trial 1: Ready to eat cereal incorporated with fibres Low-fibre (4 g) AXOS or AX from flax (FLAX) High fibre (15 g) AXOS or AX from flax (FLAX) Trial 2: Low fibre High fibre AXOS High fibre FLAX Low fibre isocaloric		Overweight Women BMI 25.0–29.9 kg/m^2^	= postprandial appetite ↗ GLP-1 ↗ PYY (High fibre AXOS and High fibre FLAX vs. Low fibre isocaloric)	[196]

### 3.8. Nutritional and Health Claims of Wheat Bran

According to the World Health Organization, the recommended intake in DF is at least 25 g of total fibre per day for the average population [10]. Nutritional claims such as “high fibre” and “source of fibre” can be used whenever fibre content reaches at least 6% or 3%, respectively [253]. Another nutritional claim can be used for mineral and vitamin content on wheat products if specific conditions of content are met. The European Commission (EC) has authorised three health claims for wheat bran products following evaluation by the European Food Safety Association (EFSA) (Table 7). They are linked to fibre hydration properties, and their effects are “increasing the faecal bulk”, “reducing intestinal transit time”, and AX effect on “reduction of postprandial glycemic response”. β-glucans from oats and barley have been associated with “reduction of blood cholesterol”, “maintenance of blood cholesterol”, and “reduction of postprandial glycemic response”, but β-glucans from wheat have not been included in these claims [254]. In the United States of America, the code of federal regulation authorises some health claims about DF on cancer and SDF on CVD. But they are associated with a specific vocabulary, such as “Diet low in fat and high in fibre-containing grain product may reduce the risk of some cancers” [255], as little scientific evidence has been provided by clinical trials. As Moshawih et al. have shown in their article, there is no approved health claims on fibre effects on the reduction of cancers, reduction of CVD, prebiotic effects and stimulation of immune responses [256]. There are no claims authorised for phenolic acid properties. This means that more clinical trials need to be done to valorise these effects in food products. 

## 4. Improving Wheat Bran Characteristics for Increasing Health Benefits

As we saw in the previous part, DF has more or less validated effects on glucose metabolism, cholesterol, intestinal transit, satiety, immunomodulatory activity, cell protection, CVD, and obesity prevention. But excessive DF intake can lead to negative effects: reduction of vitamin, mineral, protein and energy absorption, diarrhoea, flatulence, bloating, abdominal discomfort [257]. DF is known to elevate stool weight and lessen colonic transit time [258], but increasing IDF consumption can lead to abdominal pain and constipation [259]. Also, reducing or stopping DF intake can improve constipation and its associated symptoms. Fibre can be ingested as a part of the ingredients, for example, the incorporation of wheat bran in bread (whole-wheat bread), or as isolated fibre. The incorporation of fibre in bread reduces the bread glycaemic index [260]. But the addition of WB in bread causes detrimental effects on dough properties which can make the process harder, mostly due to IDF. Changes in bread loaf volume, colour, texture, and taste can occur, and they are not wanted by the consumers [261]. To improve the nutritional value of bread products without altering the rheological and olfactory characteristics, wheat bran treatments were developed. Reduction of Mw and/or improvement of SDF content and modifications of the functional properties of WB using biological, physical, chemical, or combined approaches were the main objectives. Despite all these changes, there is no concrete evidence of any difference in health effects between SDF and IDF. Both appear to be beneficial to health.

### 4.1. Biological Treatments

Biological treatments include fermentation with fungi, yeast, bacteria, and enzymatic hydrolysis (Table 8). With an increase in SDF content, the major effect of biological treatment is an increase in polyphenol content with abundant bioactivity. Higher TPC have been observed as a result of combined yeast and bacterial fermentation [49]. These treatments increase free FA release, in particular when microbial fermentation is combined with enzymatic hydrolysis [262]. A higher free soluble phenolic content is reported for wheat bran fermentation with fungi versus yeast or bacteria. Bacillus-treated wheat bran had the lowest level [263]. This is associated with higher antioxidant activity [262]. However, Tanasković et al. (2021) reported an increased EC_50_ for DPPH analysis after 8 days fermentation with *Bacillus* sp. TMF-2 even though TPC increased [167]. This is due to the capacity of *Bacillus* to further transform ferulic and *p*-coumaric acids into vanillin and *p*-hydroxybenzoic acids which have lower antioxidant activity, suggesting a species- and time-dependent response. This was confirmed by Zhang et al. [263]. Temperature positively impacts polyphenol release, so a combination of biological treatment with thermal processing is the most efficient method to release FA and, consequently, increase antioxidant properties [264]. However, a loss of TPC can result from high-temperature treatment, as phenolic acids are temperature-sensitive. Increasing alkylresorcinol content has also been noticed, due to a difference in pH during fermentation [265]. Some articles on microbial fermentation report a reduction of phytic acid content to around 20 to 37% (Table 8). Phytic acid is a molecular marker for the aleurone layer; it is the storage form of phosphorous, which is considered an antinutrient because it chelates micronutrients (Ca^2+^, Mg^2+^, Fe^2+^, Zn^2+^, amino acids). It is around 50 to 62 mg/g DW in wheat bran [142]. Phytic acid affects the textural properties of bread, so its reduction could improve the nutritional and textural profile of whole-wheat bread made with fermented bran [130]. In addition, a number of authors noticed differences in the flavour profile and aroma intensity which improved the sensorial profile of wheat bran after microbial fermentation (fungi, bacteria or bacteria and yeast) [130,153,179,266]. Spaggiari et al. detailed the volatile profile after *Lactobacillus* fermentation and observed a reduction in carboxylic acids and esters, an increase in aldehyde compounds and furan derivatives which, for the latter, are characteristics of bread aroma [179]. The alcohol class was the most represented, and ketone abundance was variable. Another study on lactic acid bacteria fermentation combined with extrusion has shown an increase in volatile compounds, leading to a better overall acceptability by the consumers [267]. The study authors reported that this treatment did not increase dough hardness. 

Yeast fermentation and enzymatic hydrolysis with xylanase increased SDF and WEAX content [25]. Increasing of WEAX has been reported in other articles for bacterial and yeast combined with bacterial fermentations [153,179]. Higher SDF content is often associated with higher binding and hydration capacities. Probiotic fermentation of IDF led to increased WRC, OBC, SCAC, and NIAC but reduced CAC [73]. SCAC decreased with particle size reduction and enzymolysis, which can be linked with IDF reduction, as we discussed in the SCAC section [142]. Liu et al. (2021) observed significant modification of physico-chemical and functional characteristics of wheat bran modified by snail enzymes [156]. The authors showed increased antioxidant capacity and better GAC and CAC. Although snail enzyme is low-cost and easily accessible, the protocol seems to be long and complicated because of the enzyme extraction process and the multi-step treatment under specific conditions. This is potentially hard to scale-up for an industrial goal. Enzymatic treatment can be coupled with physical treatment to improve both solubility and hydration properties.

### 4.2. Physical Treatments

Physical treatments include milling, microfluidisation, micronisation, steam treatments, and extrusions. They are ways to reduce the particle size, generally producing fractions corresponding to superfine (>25 µm), fine (25–200 µm), medium (200–500 µm), and coarse (500–1000 µm). Fractions can then be separated by sieving or electrostatic separation. This can be useful for the extraction of the aleurone layer, for example [269,270].

In the literature, we observe that the reduction of WB particle size by physical treatment reduces IDF content and increases SDF content (Table 9). Authors shows that stream explosion, superheated steam and extrusion increase fibre solubility (Table 9). Insoluble AX are transformed into WEAX and new water-soluble indigestible glucans are synthetised through transglycosidation by blasting extrusion [59]. 

WEAX, β-glucans DP, and increased damaged starch proportions will improve viscosity, leading to improved binding properties. Microfluidisation, extrusion, and steam explosion significantly improve wheat bran WHC and WSC. Microfluidisation leads to an expansion of particles and increases the porosity. The extrusion process is a combination of thermic and mechanic treatments using different shear rates. The screw speed has a bad influence on WBC, but it has a good effect on WSC because it will increase the WSC with high screw speed [56]. Ralet et al. (1990) noticed no significant difference on WBC after extrusion-cooking but they observed that low treatment intensity increased water uptake while severe conditions slowed it down [61]. This can be explained, since medium conditions can create pores, which increases the water uptake, and drastic treatment can induce collapse, which limits the water uptake. Indeed, for milling and micronisation processes, a reduction of hydration properties is reported. These processes break the fibre matrix and collapse pores. For example, micronisation yields a loss of micropores, which reduces the WRC [271]. Ye et al. (2021) defined extrusion as the greatest improvement in the nutritional and processing qualities of bran compared to five other WB modification treatments [25]. That conclusion was confirmed by Lee et al. in 2021 [60]. General effects of extrusion are: (i) antinutritional factor destructuring; (ii) gelatinisation of starch; (iii) increasing of SDF content; and (iv) reduction of lipid oxidation. This process is low-cost, has high productivity and energy efficiency and has been used since 1856 for food processing technology [272]. Extrusion increases folic acid, SDF, free phenolic (+17.6%) and WEAX (+33.6%) contents, WSC (1.25 times that of control), WHC and ORAC, but it reduces vitamins B1 and B2 [25]. Extrusion can also reduce compounds, which can limit DF, omega-3 fatty acids, starch, proteins, minerals and vitamin use, including phytic acid. Under extrusion and steam treatments, phytic acid content decreases, possibly a consequence of the high temperatures used [142]. The high-temperature pressure during extrusion causes starch gelatinisation and degradation, denaturation of some proteins, and creation of complexes between starch and protein; all of this decreases fat, starch and protein content [25]. 

The particle size reduction and the breaking of the aleurone layer increase the exchange surface of wheat bran and the bioactive compound bioaccessibility, which can be 65% higher in superfine wheat bran [58]. For scavenging capacities, milled WB has higher values for ORAC, DPPH and ABTS [58,175]. It should be noted that better reduction capacities have been observed for coarse wheat bran (>900 µm) compared to fine wheat bran (≈200 µm) for FRAP and total antioxidant capacity [175], meaning that antioxidant capacity depends of the method used. The reduction of particle size increases surface area, which improves the extraction of phytochemicals, affects polyphenol content and antioxidant activity. Indeed, very high free phenolic acid, flavonoid, and polyphenol pigment (anthocyanin and carotenoid) contents have been observed [175]. Li et al. confirmed in 2022 that phenolic compounds are directly impacted by particle size changes, especially *p*-coumaric acid, which is five times higher due to complete aleurone destruction [58]. Polyphenol release will directly increase, in particular FA. It should be noted that a combination of thermal processing with milling presented inverted effects, with a negative impact on FA content and antioxidant activity in wheat [262]. We saw that FA have the capacity to inhibit starch digestive enzymes, by interacting with amylopectin to retard starch digestion; that is probably why a decrease in starch digestibility has been observed when phenolic compound release increases [58]. Indeed, Li et al. observed reduced carbohydrate digestive enzyme (α-amylase and α-glucosidase) activities on superfine bran and reduced starch digestibility [58]. Some researchers who studied particle-size reduction process effects on starch report increasing damage of starch due to the impact force and collisions between particles. This point is interesting to potentially lower post-prandial blood glucose response. 

Decreasing particle size would have an influence on increasing SCFA synthesis because it improves DF fermentability [273]. Higher propionic and butyric content has been observed for 90 µm WB in vitro digestion than in 500 µm bran [274]. Butyric acid enhances the inflammatory response with higher quantities of immunoglobulin A (IgA) secreted in powdered WB [273]. 

Modified WB particle size is an easy way to increase solubility and bioactive compound bioavailability. Physical dry treatments, which are different from wet processes (chemical or biological), do not need water or require a stabilisation process and are easily scalable [275]. In addition, they improve some health-related parameters: solubility, viscosity, scavenging antioxidant capacity, and lower starch digestion, depending on the process and conditions. But these processes with high-energy impact and heat present drawbacks. These conditions will deeply impact WB through breakage, thus producing negative effects on hydration properties, dough quality and bread making, and on sensitive nutrients such as vitamins. Steam pressure-cooking reduces the fibre Mw and viscosity but it also reduces the fibre’s ability to attenuate plasma glucose and insulin concentrations [196]. For example, steam explosion treatment on bran is detrimental for bread quality due to the protein denaturation or Maillard reactions between reducing sugars and amino acids [276].

**Table 9 foods-12-02693-t009:** Examples of wheat bran modifications with physical treatments and their effects on functional, rheological and physico-chemical characteristics. “↗” is for increase and “↘” for decrease.

Treatment Type	Objective	Effects of Particle Size Reduction	References
Milling	To study if particle size distribution of whole wheat bran affects the phytochemical extractability and antioxidant properties.	↗ Phenolic acids, flavonoid, carotenoids, anthocyanins ↗ ORAC (+80%) ↘ DPPH, FRAP, TAOC	[175]
Milling	To understand wheat bran’s hydration properties.	No chemical modification ↗ WEAX ↗ Damaged starch ↘ WHC	[152]
Milling	To study the effect of wheat bran particle size on in vitro gastrointestinal digestion.	↗ Release of Phenolics compounds ↗ Antioxidant capacities ↗ Carbohydrate-digestive enzymes inhibitory activities ↘ Starch digestibility (retard starch digestion)	[58]
Air-flow impact mill	To study the link between structure and hydration properties of milled wheat bran.	↗ SDF ↘ WRC, WSC, ORC	[86]
Micronisation = superfine grinding	To investigate the effect of wheat bran micronisation on its functionality and dough properties.	↗ SDF ↗ WEAX ↗ Damage starch ↗ Phenolic acids liberation ↗ Free ferulic acid ↗ TPC, ABTS, DPPH ↗ Aleurone cells disruption ↘ Dough extensibility ↘ IDF ↘ WRC	[271]
Microfluidisation		↗ WEAX ↗ Extractible B-glucans ↗ Viscosity ↗ Water extractability ↗ WRC ↗ Phytate (aleurone breaking) ↗ Free glucose content ↗ Maltose (starch degradation) ↗ Water extractible proteins	[112]
Microfluidisation	Improve bran’s physiological properties	↗ WHC, WSC ↗ OHC ↗ CEC ↘ Bulk density	[277]
Microfluidisation	Improving antioxidant properties of bound phenolic compounds.	↗ Surface reactive (×3.8) and bound phenolics ↗ Antioxidant capacities	[278]
Extrusion		↗ Fibre solubility ↘ Total dietary fibre ↘ WBC ↘ Phytic acid content	[56]
Extrusion		↘ Phytic acid (64.4%) Ture 115 °C 20% moisture ↘ Polyphenols ↘ Oxalates (36.8%, 140 °C, 20% moisture) ↘ Trypsin inhibitors (71.2%, 140 °C, 20% moisture) ↗ Bulk density ↗ Redness ↘ Yellowness	[279]
Extrusion		↗ Soluble fibre (70–100% RPM max)	[280]
Extrusion		↗ SDF ↗ WEAX (+33.6%) ↗ WHC, WSC ↗ Total phenolic ↗ FA ↗ Folic acid ↗ ORAC ↘ OHC ↘ Vitamin B1/B2 ↘ Phytic acid ↘ Fat, starch, protein	[25]
Blasting-extrusion		↗ SDF (+70%) ↗ WRC, WSC ↗ OBC ↗ Water soluble polysaccharides ↗ Antioxidant	[28]
CO_2_ Blasting-extrusion		↗ SDF ↗ WRC, WSC ↗ ORC	[59]
Steam explosion (215 °C 120 s)		↗ Free soluble phenolic acids	[281]
Steam explosion		↗ WEAX ↗ Reducing sugar ↘ Baking quality ↘ Phytic acid	[276]
Steam explosion (high-pressure steam 0.8 MPa, 5 min)	Improve the value of wheat bran to provide a reference for the development of WB treatment	↘ Particle size ↗ WHC, WSC (↘ WHC with particle size reduction) ↘ SCAC (and with particle size reduction) ↘ CEC ↘ Phytate content	[142]

### 4.3. Chemical Treatments

Chemical treatments are used on wheat bran to extract fibres and their bioactive compounds or to enhance WB functional properties. They include acid, alkali hydrolysis, etherification, esterification, carboxymethylation, hydroxypropylation, cross-linking, and oxidation treatments. 

Acid treatment can improve rheological properties, but when acid is too concentrated, it can alter hydration and the functional properties of wheat bran (GAC, SCAC, CAC). Alkali treatment increases SDF and improves bile acid-binding capacity, WHC, α-amylase activity, GAC, WSC, and thermal stability. A coupled acid-alkaline treatment had the most significant results on improving fibre solubility [24,282,283].

Carboxymethylation is the introduction of carboxymethyl groups into the polysaccharide chain; it has a destructuring effect on cellulose and hemicellulose. This effect is due to the etherification of hemicellulose and the cellulose hydroxyl group by a carboxymethyl group, which increases hydrophilic properties [284]. Destructuration creates a honeycomb structure of IDF and higher specific surface area, which enhance oil retention capacity (ORC). GAC is improved by increasing viscosity, porosity, and specific surface area. An increase in GAC is related to better reduction of blood glucose levels. NIAC prevents nitrite toxicity and is linked to ferulic acid. NIAC is reduced by carboxymethylation which might lead to a loss in FA. Even so, upgraded antioxidant capacities and TPC are found [285]. The effect of carboxymethylation on polysaccharides from other species shows in vitro antioxidant, antitumor activity, immunoregulatory activity, and antimicrobial activity, but there is not enough data on wheat bran polysaccharides to expand these potential effects [286]. Hydroxypropylation on barley fibre improves their digestibility, but less than carboxymethylation. Carboxymethylation and hydroxypropylation are complicated treatment process with many steps [24].

Subcritical pretreatments (hydrothermal, pressurised hot water, or autohydrolysis) are treatments using water, elevated temperatures, and pressure. The optimal conditions are between 120 to 180 °C for 10 to 40 min to extract bioactive compounds [287]. These treatments are used to solubilise hemicellulose mostly for biorefinery purposes or uses. The traditional extraction solvents used are water, alkaline or acidic solutions. Subcritical water extraction (SWE) is a green treatment used to extract SDF from wheat bran. A combination of SWE with ultrasound could convert some IDF into SDF but, for instance, there is no evidence of this phenomenon, and only an increase in the yield of SDF extraction has been observed [139]. SWE coupled with ultrasound can cause Mw reduction and enhance viscous properties but does not increase phenolic content. However, antioxidant activity with FRAP was improved. This may be related to higher uronic acid and reducing sugar contents [139]. Indeed, the antioxidant power is influenced by monosaccharide composition, glycosidic linkages of polysaccharides, and its degree of substitution and DP [288,289]. SWE can be coupled with alkali, acid hydrolysis or ozonolysis treatment to improve enzymatic digestion and hemicellulose solubilisation. The addition of citric acid treatment increases SDF dissolution thanks to carboxyl groups (–COOH) that can ionise hydrogen ions (H^+^); it also increases reducing sugar and uronic acid contents, which can improve biological activities. Incidentally, lower Mw and apparent viscosity induced stronger inhibitory effects on α-amylase. SWE with citric acid highly disrupt cell wall structures, resulting in a larger surface area. According to Li et al. (2021), ozone treatment destructures lignin and forms acids, in particular oxalic acids; it increases acidity and enhances hemicellulose solubility by the destructuration of wheat bran [290]. 

The ozonation process is an oxidative treatment used for whitening fibre and improving flavours because it oxidises pigments and transforms ferulic acid into vanillin [291,292]. Indeed, polyphenols are very reactive to ozone. Ozone action on dietary fibre starts with lignin, which is preferentially targeted by ozone due to its high electron density regions (double bonds) and aromatic rings. Ozone will interact with lignin aromatic compounds and double bonds and oxidise alcohol hydroxyl, alkyl ether, and aryl groups into carbonyl groups, and aldehyde groups into carboxyl groups. It is for these reasons that ozone is already used for delignification [292]. Then, ozone will break phenolic crosslinks between NSP (hemicellulose/hemicellulose and hemicellulose/lignin cleavages), resulting in increased SDF contents. Ozone is able to depolymerise fibre because a reduction of β-glucans Mw after ozone treatment has been reported [79]. This increase in SDF in wheat bran may imply the transformation of AX in WEAX; Li et al. in 2021 reports increasing hemicellulose solubility by ozonolysis treatment [290]. Indeed, ozone is capable of reducing the cellulose degree of polymerisation by glycosidic bond cleavage (direct cleavage at C1 position or after decarboxylation process) and oxidation of hydroxyl groups [293]. However, it should be a low effect, because the decarboxylation process is a long three-step process unfavoured by ozone. This chemical treatment has an effect on hemicellulose solubility, so it could enhance health benefits via a green process. In France, during 2003 and 2004, AFSSA rendered two opinions regarding the safety of using ozone as an auxiliary technology for wheat grain treatment before milling. The use of ozone has been authorised by the regulatory agency since 2006 as a processing aid for flour quality improvement, based on treatment by ozone in a closed sequential batch reactor [294]. Some articles studying the effect of ozone on food cereal products made from ozonated grain and a moderated ozone treatment presented enhanced rheological properties [295]. No data are available for the nutritional and health aspects. That is why more studies should be done on ozone effects on wheat bran NSP to understand the mechanisms of ozone action and to study potential nutritional improvement. 

Changing the structure of wheat bran directly impacts physicochemical and functional properties. The increase in SDF content can upgrade WB’s functional and nutritional properties, but it can also alter rheological properties, making the product unwanted by the consumer. That is why alternative protocols have been optimised. Among them, enzymatic methods are interesting for the polyphenol bioavailability and flavour enhancement, but are expensive in terms of enzyme cost and long reaction time (>24 h); specific conditions are needed, and microorganism control is not easy. In addition, the improvement of binding capacities with biological methods seems to be less efficient than physical treatment [142]. Physical treatments are cheaper and easier to scale up but hard conditions greatly impact the final product. Chemical treatments result in high water consumption, corrosiveness, and are not environmentally friendly. Not all are adapted for food use, but can be interesting for the extraction of a specific fibre. Ozone treatment is not currently used but can be interesting because it has positive effects on SDF content and flavour profiles. It is a green treatment considered as Generally Recognized As Safe (G.R.A.S), easy to scale up, without residue, and does not destroy the matrix as physical treatments do. 

## 5. Conclusions

Wheat bran is rich in dietary fibre and is known to have beneficial effects on health, in particular on intestinal diseases management and prevention of cardiovascular diseases, cancer, obesity, and type-2 diabetes. The effects of dietary fibres depend on the molecular composition and the complex structure. Due to WB component functional properties (viscosity, hydration properties, etc.) that have negative impacts on food processing, researchers and industrials try to improve wheat bran dietary fibre by modifying their structures and functional properties. But the processes can alter the rheological properties and the nutritional value. These unwanted alterations, the high cost and environmental impact of these techniques leave room for better procedures. In this review, we saw that functional properties are strongly correlated with non-starch polysaccharide structure and physico-chemical composition. Wheat bran health effects, however, still lack sufficient clinical proof and regulatory validation in the form of health claims. Despite all, a lot of modifying processes succeed in improving soluble dietary fibre content without showing clinical evidence that wheat bran structure modifications really enhance health benefits. The current knowledge on structure–activity relationships can lead to process innovation for more soluble and bio-efficient components of WB. They will then need to be tested in clinical trials before generic or innovative health claims can be proposed to the consumers.

## Figures and Tables

**Figure 2 foods-12-02693-f002:**
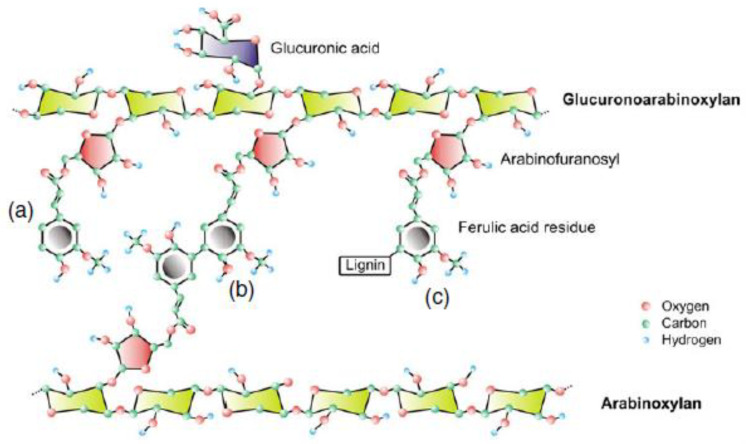
Feruloylated glucuronoarabinoxylan (FA-GAX): (**a**) ferulic acid residue esterified to the arabinofuranosyl residue of GAX; (**b**) diferulic acid cross-linking two FA-GAX; (**c**) ferulic acid residue anchoring lignin to the GAX [46].

**Figure 3 foods-12-02693-f003:**
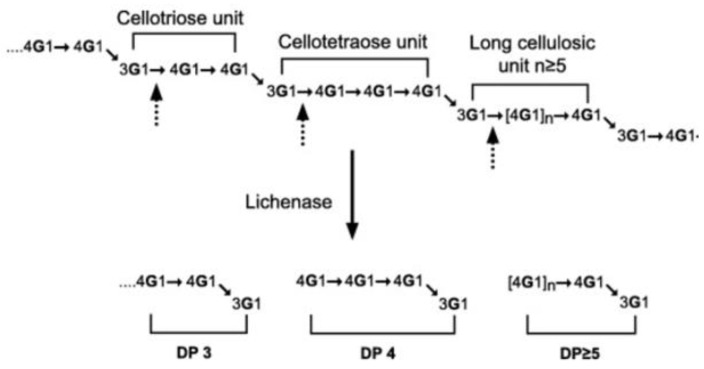
Structure of β-glucan before (**up**) and after (**down**) lichenase digestion. Polysaccharide → oligosaccharides [19]. A cellotriose unit is a trisaccharide (DP3); A cellotetraose unit is a tetrasaccharide (DP4); and a long cellulosic unit is an oligosaccharide (DP > 5) [50].

**Figure 4 foods-12-02693-f004:**
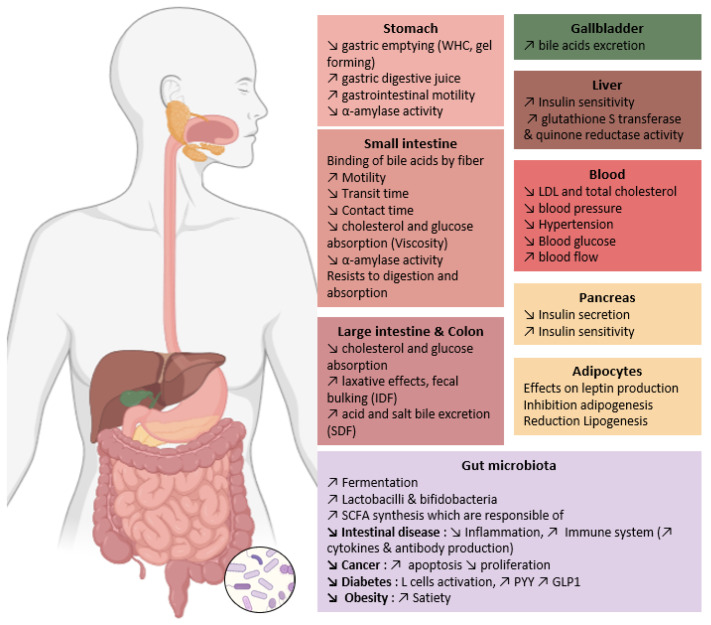
Summary of dietary fibre from wheat bran biological effects (Made with Biorender.com, Scientific Image and Illustration Software|BioRender accessed on 12 July 2023). ↗ is for increase and ↘ for decrease.

**Table 7 foods-12-02693-t007:** European authorised health claims on wheat fibre (https://ec.europa.eu/food/safety/labelling_nutrition/claims/register/public/?event=search (accessed on 22 November 2022).

Nutrient, Substance, Food	Claim	Conditions	Health Relationship	EFSA Opinion Reference/Journal Reference	Commission Regulation
Wheat bran fibre	Wheat bran fibre contributes to an increase in faecal bulk	The claim may be used only for food which is high in that fibre as referred to in the claim HIGH FIBRE as listed in the Annex to Regulation (EC) No 1924/2006	Increase in faecal bulk	2010;8(10):1817	Commission Regulation (EU) 432/2012 of 16 May 2012
Wheat bran fibre	Wheat bran fibre contributes to an acceleration of intestinal transit	The claim may be used only for food which is high in that fibre as referred to in the claim HIGH FIBRE as listed in the Annex to Regulation (EC) No 1924/2006. In order to bear the claim, information shall be given to the consumer that the claimed effect is obtained with a daily intake of at least 10 g of wheat bran fibre	Reduction in intestinal transit time	2010;8(10):1817	Commission Regulation (EU) 432/2012 of 16 May 2012
Arabinoxylan produced from wheat endosperm	Consumption of arabinoxylan as a part of a meal contributes to a reduction of the blood glucose rise after that meal	The claim may be used only for food which contains at least 8 g of arabinoxylan (AX)-rich fibre produced from wheat endosperm (at least 60% AX by weight) per 100 g of available carbohydrates in a quantified portion as part of the meal. In order to bear the claim, information shall be given to the consumer that the beneficial effect is obtained by consuming the arabinoxylan (AX)-rich fibre produced from wheat endosperm as part of the meal.	Reduction of post-prandial glycemic responses	2011;9(6):2205	Commission Regulation (EU) 432/2012 of 16 May 2012

**Table 8 foods-12-02693-t008:** Some examples of wheat bran modifications with biological treatments and their effects on functional and physico-chemical characteristics. “↗” is for increase, “↘” for decrease and “=” for no change.

Microorganism/Enzyme	Matrix	Objective and Perspectives	Effects in Comparison with Control Sample	References
*Eurotium cristatum* (Fungi)	Wheat bran	To study if *E. cristatum* has a potential to produce wheat bran food products more nutritional, flavourful and functional.	↗ Ferulic acid content ↗ SDF ↗ Bindings capacities (WSC, WHC, OBC) ↗ TPC, anthocyanins, phenolic acids ↗ DPPH and ABTS ↗ Pancreatic lipase inhibition activity ↗ phenylethyl alcohol Different flavour	[130]
*Fomitopsis pinicola* (Fungi)	Wheat bran	Evaluate the potential application of *F. pinicola* to improve the physicochemical and functional properties of wheat bran.	↗ Polyphenols ↗ Alkylresorcinols ↗ AO activity ↗ Swelling capacity ↗ Protein ↘ Phytic acid	[265]
*Lactobacillus plantarum 423* (Bacteria)	Rice bran and wheat bran	To study the potential application of rice and wheat bran in health foods and nutraceuticals.	↗ odour intensity ↗ AO activity	[266]
*Bacillus* species, yeasts, filamentous fungi (Bacteria, fungi, yeast)	Wheat bran	Improving phenolic acid composition and antioxidant activity of wheat bran.	↗ free phenolic content ↗ DPPH AO activity correlated with TPC	[263]
*Bacillus* sp. TMF-2 (strain which produces several enzymes)	Wheat bran	To produce wheat bran with higher nutritional quality.	↗ Total phenolic content (×3) ↗ Antioxidant capacity (FRAP ×10) ↗ hydrolytic enzymes ↘ phytic acid	[167]
*Lactobacillus rhamnosus* (Bacteria)	Wheat bran	To study the ability of lactic acid bacteria to modify the overall characteristics of wheat bran as a pre-treatment to potentially enhance its health and sensory properties.	↗ WEAX ↗ Fruity note at volatile profile ↘ 37% phytic acid	[179]
*L. plantarum* (Mixed acid lactic bacteria)	Wheat bran IDF	To study structure, physiochemical, functional properties and antioxidant activity of wheat bran modified IDF by fermentation in order to provide high-quality functional IDF for food processing in human health management.	↗ WRC, Oil retention Capacity (ORC), WSC ↗ NIAC, TPC ↘ CAC	[73]
*L. acidophilus* (Bacteria)			↗ SCAC CEC ↗ NIAC, TPC ↘ CAC	
Yeast and lactic acid bacteria	Wheat bran	To improve the nutritional, physical and flavour properties of wheat bran.	↗ WEAX ↗ SDF ↗ Alkylresorcinols ↗ Binding/hydration capacities ↗ Phenolic content ≠ flavour ↘ 20% phytic acid	[153]
*Baker’s yeast*	Wheat bran	To compare different treatments and evaluate their effects on wheat bran properties in order to improve its quality.	↗ WEAX (+46.4%) ↗ SDF ↗ Free Phenolic content ↗ WHC ↗ Vitamin B2 ↘ Vitamin B1 ↘ OHC	[25]
Enzymatic treatment with xylanase	Wheat bran		↗ Folic acid ↘ Vitamin C, B1, B2 ↘ Phytic acid ↗ Bound phenolic content ↘ Free phenolic content↘ DPPH scavenging activity (−16.9%) ↘ WHC, OHC	[25]
Enzymatic hydrolysis with cellulase and xylanase (1:3)	Wheat bran	Improve the value of wheat bran to provide a reference for the development of WB treatment	= WHC (↘ WHC with particle size reduction) ↘ SCAC (and with particle size reduction) ↘ CEC = Phytate content	[142]
Snail enzymes	Wheat bran	To modify IDF and SDF from wheat bran in order to improve their functional and physico-chemical properties for potential application in the food industry as a functional ingredient.	↗ IDF Oil retention capacity ↗ Glucose adsorption ↗ Cholesterol adsorption ↗ Radical scavenging activity (DPPH) Change in microstructure (SEM) Hemicellulose and cellulose degradation	[156]
Hydrothermal and enzymatic treatment	Wheat bran	To investigate a wheat bran pre-treatment for it use as a feedstock for biorefineries.	Sugars dissolution Transformation of carbohydrates in free sugars Partial dissolution of hemicellulose	[268]

AO = Antioxidant.

## Data Availability

Not applicable.

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
