# Peer review of "Functional and Nutritional Characteristics of Natural or Modified Wheat Bran Non-Starch Polysaccharides: A Literature Review"

_foods, 2023, doi:10.3390/foods12142693_

Round 1

Reviewer 1 Report

Review of manuscript Functional and nutritional characteristics of natural or modified wheat bran non-starch polysaccharides: a literature review. Wheat fibres

With the large and growing human consumption of wheat – the topic of the literature review is a relevant one. The review highlights the beneficial health-related for effects of soluble fibres compared to insoluble fibres. This aim and the way it is described in this review is questionable for several reasons

1.      The differentiation soluble-insoluble for health benefits is not correct and should be phased out  – according to EFSA, WHO and ref. 16. As stated by the World Health Organisation and EFSA (2010) doi: 10.2903/j.efsa.2010.1462  and mentioned also in Ref 16      The terms “soluble” and “insoluble” dietary fibre have been used in the literature to differentiate between viscous, soluble types of fibre (e.g. pectins) and insoluble components such as cellulose. (…)However, this differentiation is method-dependent, and solubility does not always predict physiological effects. Therefore, FAO/WHO proposed the distinction between soluble and insoluble fibre should be phased out (FAO/WHO, 1998).

2.      In lines 44-50 the statement is made that beneficial health effects of fibres are mostly linked to soluble fibre content   Ref 16 is cited. However, Ref 16 – an authoritative review - is stating in several paragraphs the contrary - both soluble and insoluble fibre are important for health benefits

3.      Low molecular weight fibres (DP 3-9) are an important part of soluble fibres. Therefore it is an omission that in the section on analytical methods only old AOAC methods for Total Dietary Fibre are mentioned – such as methods 985.29 and 991.43 – that don’t measure fibres with DP 3-9. The new AOAC methods should be mentioned instead: AOAC 2009.01 and 2022.25 – the ones measuring TDF according to the fibre definitions of de FDA and the EU.

4.      As mentioned in 2. References are used incorrectly, to ‘substantiate’ the importance of soluble fibres. Also, relevant reviews highlighting the importance of the insoluble wheat bran are not included. Such as Jefferson A and Adolphus K (2019) The Effects of Intact Cereal Grain Fibers, Including Wheat Bran on the Gut Microbiota Composition of Healthy Adults (2019) doi: 10.3389/fnut.2019.00033

These points indicate the limited overview of the authors of the world of dietary fibre. This is also illustrated by lines 126 and 127 where AACC  and AOAC are incorrectly mentioned American Association for Clinical Chemistry and Association of Official Agricultural Chemists instead of American Association for Cereal Chemistry and Association of Official Analyticl Chemists.

I strongly advise you to include a dietary fibre expert as co-author or senior adviser, when you write a review paper on dietary fibres 

No comments

Author Response

Dear reviewer,

Thank you for your valuable comments that have notably improved the quality of our manuscript.

Best regards

Reviewer 2 Report

Congratulations on your excellent review on "Functional and nutritional characteristics of natural or modified wheat bran non-starch polysaccharides." The review of the literature is very complete and detailed, since it ranges from the anatomical conformation of the wheat grain in which its different parts are shown, as well as the layers that make it up and the different compounds that are present in each of them.

The different extraction techniques and conditions of the different compounds are also mentioned, as well as the different methods that are used for their identification and quantification of each one of these compounds. Much emphasis is placed on the effects on health, as well as on the control and prevention of cardiovascular diseases, cholesterol, obesity, type 2 diabetes and cancer. It is concluded that the benefits provided by non-starch posaccharides, including arabinoxylans and b-glucans, depend on the dose and their molecular characteristics, including the concentration, solubility, viscosity, molecular weight and bioavailability of the linked polyphenols. . There is also talk of the existence of physical, chemical, biological methods, as well as combined methods to modify the molecular characteristics of non-alminous polysaccharides. Most of these techniques aim to increase the soluble dietary fiber content in order to potentially improve its nutritional and health benefits. The chemical impact on the structure of these compounds is well known, what is necessary is to determine the biological effects due to these structural changes.

Author Response

(The authors gave the same response as above.)

Reviewer 3 Report

General comments

The proposed review manuscript lacks novelty which makes its added value to the scientific community very limited. The functionality and nutritional properties of wheat bran fibers have recently (2020 – 2023) been reviewed multiple times:

Functional properties

Maina, N.H.; Rieder, A.; De Bondt, Y.; Mäkelä-Salmi, N.; Sahlstrøm, S.; Mattila, O.; Lamothe, L.M.; Nyström, L.; Courtin, C.M.; Katina, K.; Poutanen, K. Process-Induced Changes in the Quantity and Characteristics of Grain Dietary Fiber. Foods 202110, 2566. https://doi.org/10.3390/foods10112566

Gan, Jiapan, et al. "Systematic review on modification methods of dietary fiber." Food Hydrocolloids 119 (2021): 106872.

Onipe, O.O.; Ramashia, S.E.; Jideani, A.I.O. Wheat Bran Modifications for Enhanced Nutrition and Functionality in Selected Food Products. Molecules 202126, 3918. https://doi.org/10.3390/molecules26133918

Pietiäinen, Solja, et al. "Effect of physicochemical properties, pre-processing, and extraction on the functionality of wheat bran arabinoxylans in breadmaking–A review." Food Chemistry (2022): 132584.

Nutritional properties

Wen Cheng, Yujie Sun, Mingcong Fan, Yan Li, Li Wang & Haifeng Qian (2022) Wheat bran, as the resource of dietary fiber: a review, Critical Reviews in Food Science and Nutrition, 62:26, 7269-7281, DOI: 10.1080/10408398.2021.1913399

Lise Deroover, Yaxin Tie, Joran Verspreet, Christophe M. Courtin & Kristin Verbeke (2020) Modifying wheat bran to improve its health benefits, Critical Reviews in Food Science and Nutrition, 60:7, 1104-1122, DOI: 10.1080/10408398.2018.1558394

Yao, Wanzi, et al. "The effects of dietary fibers from rice bran and wheat bran on gut microbiota: An overview." Food Chemistry: X (2022): 100252.

Nirmala Prasadi, V.P.; Joye, I.J. Dietary Fibre from Whole Grains and Their Benefits on Metabolic Health. Nutrients 2020, 12, 3045. https://doi.org/10.3390/nu12103045

Baoshi Wang, Guangyao Li, Linbo Li, Mingxia Zhang, Tianyou Yang, Zhichao Xu & Tengfei Qin (2022) Novel processing strategies to enhance the bioaccessibility and bioavailability of functional components in wheat bran, Critical Reviews in Food Science and Nutrition, DOI: 10.1080/10408398.2022.2129582

Saini, Praveen, et al. "Wheat Bran as Potential Source of Dietary Fiber: Prospects and Challenges." Journal of Food Composition and Analysis (2022): 105030.

Liu, Jie, Liangli Lucy Yu, and Yanbei Wu. "Bioactive components and health beneficial properties of whole wheat foods." Journal of agricultural and food chemistry 68.46 (2020): 12904-12915.

A minor remark is that the authors did not discuss the non-starch carbohydrate fructan.

Author Response

(The authors gave the same response as above.)

Round 2

Reviewer 1 Report

Important improvements have been made in this revised version of this very long (50 pages) paper. However, I still found incorrect, non-optimal or poorly referenced statements in the introductory and definition parts. This part as such is of rather low quality and better descriptions are available elsewhere (for instance Stephen et al.). It has been a huge task to review such a broad field – with the risk that some parts may be good but others are of low quality – and better sources of information may be found in the many more specific reviews. For example, one important 100% incorrect statement is that the old AOAC methods don’t measure resistant starch. It is impossible for me to check the full paper on such incorrect statements .

Line 41  Dietary fibre is the edible portion of plants or analogous carbohydrates that are resistant to digestion and adsorption in the human small intestine but are completely or partially fer-43 mented in the large intestine by the gut microbiota [5]. A number of fibres are very poorly fermented and some not at all. Pls rephrase the 2nd part of the sentence and use a better, more recent reference (Stephen et al.

Line 49 The dietary fiber definition has never stopped evolving.  

This suggests that major changes are still occurring. This is not correct. The text in Stephen (2017) describes the reality better:  Over the years, the definition of dietary fibre has been subject to much discussion. The most recent definitions, from about 2008 (for example, Codex Alimentarius Alinorm)(2), have general global agreement. Dietary fibre is made up of carbohydrate polymers with three or more monomeric units (MU), which are neither digested nor absorbed in the human intestine

Line 54 European Food Safety Authority uses AOAC fibre as the reference for intake recommendations [9]..

This reflects the old discussion for including (like AOAC) or not (Englyst)  items such as resistant starch and low MW fibres

 In [9]an additional remark is made:  Member States are responsible to ensure conformity with the fibre definition as a whole”. Only the more recent AOAC methods such as AOAC 2009.01 and 2011.25 measure according to the current definition.

Line 62: Add sensory: dough-making properties and end product SENSORY qualities.

Line 70 In addition, nutritional claims such as “rich in fibre”. Should be “high fibre” or “high in fibre”. This is the official term used in the EU Regulation

Line 79 Over the past two decades, an important increase in consumer awareness of the health benefits of dietary fiber has been observed. Please add a reference or delete the sentence.

Line 238 AOAC 991.43 and 985.29 (Prosky /Lee) methods do not include RESISTANT STARCH and non-digestible oligosaccharides. THIS is not correct. Pls. see the figure (made by BarryMcCleary) below.

English is rather poor. Revision by a native speaker is recommended

Author Response

Dear Reviewer,

Please find attached our responses

Best regards
